# XAGen: 3D Expressive Human Avatars Generation

**Zhongcong Xu**
Show Lab
National University of Singapore
zhongcongxu@u.nus.edu

**Jianfeng Zhang**
ByteDance
jianfengzhang@bytedance.com

**Jun Hao Liew**
ByteDance
junhao.liew@bytedance.com

**Jiashi Feng**
ByteDance
jshfeng@bytedance.com

**Mike Zheng Shou** *
Show Lab
National University of Singapore
mike.zheng.shou@gmail.com

## Abstract

Recent advances in 3D-aware GAN models have enabled the generation of re-
alistic and controllable human body images. However, existing methods focus
on the control of major body joints, neglecting the manipulation of expressive
attributes, such as facial expressions, jaw poses, hand poses, and so on. In this
work, we present XAGen, the first 3D generative model for human avatars capa-
ble of expressive control over body, face, and hands. To enhance the fidelity of
small-scale regions like face and hands, we devise a multi-scale and multi-part 3D
representation that models fine details. Based on this representation, we propose a
multi-part rendering technique that disentangles the synthesis of body, face, and
hands to ease model training and enhance geometric quality. Furthermore, we
design multi-part discriminators that evaluate the quality of the generated avatars
with respect to their appearance and fine-grained control capabilities. Experiments
show that XAGen surpasses state-of-the-art methods in terms of realism, diver-
sity, and expressive control abilities. Code and data will be made available at
https://showlab.github.io/xagen.

## 1  Introduction

3D avatars present an opportunity to create experiences that are exceptionally authentic and immersive
in telepresence [10], augmented reality (AR) [22], and virtual reality (VR) [50]. These applications [1,
52, 3, 35] require the capture of human expressiveness, including poses, gestures, expressions, and
others, to enable photo-realistic generation [65, 70], animation [56], and interaction [33] in virtual
environments.

Traditional methods [11, 60, 4, 20, 23] typically create virtual avatars based on template registration or
expensive multi-camera light stages in well-controlled environments. Recent efforts [69, 43, 5, 26, 16]
have explored the use of generative models to produce 3D human bodies and clothing based on input
parameters, such as SMPL [38], without the need of 3D supervision. Despite these advancements,
current approaches are limited in their ability to handle expressive attributes of the human body, such

---

*Corresponding author

37th Conference on Neural Information Processing Systems (NeurIPS 2023).

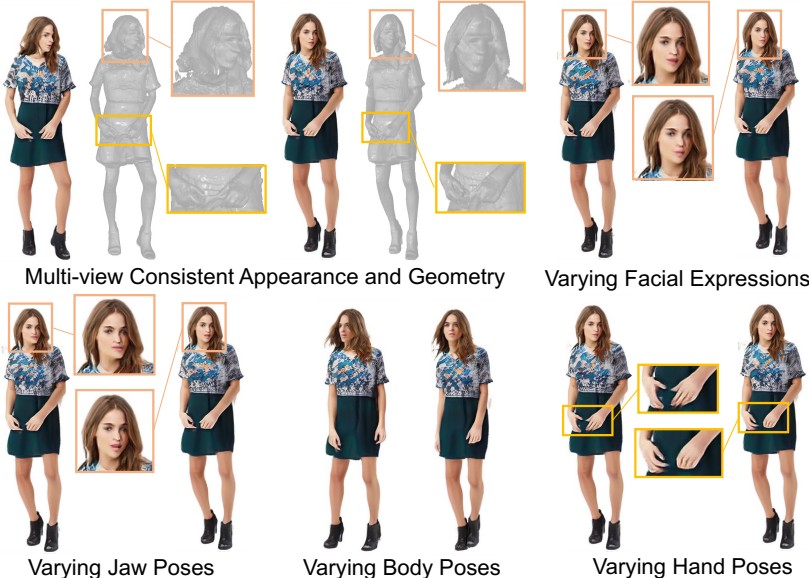

Multi-view Consistent Appearance and Geometry    Varying Facial Expressions

Varying Jaw Poses    Varying Body Poses    Varying Hand Poses

Figure 1: XAGen can synthesize realistic 3D avatars with detailed geometry, while providing disentangled control over expressive attributes, *i.e.*, facial expressions, jaw, body, and hand poses.

as facial expressions and hand poses, as they primarily focus on body pose and shape conditions. Yet, there exist scenarios where fine-grained control ability is strongly desired, *e.g.*, performing social interactions with non-verbal body languages in Metaverse, or driving digital characters to talk with various expressions and gestures, *etc.* Due to the lack of comprehensive modeling of the full human body, existing approaches [43, 5, 26] fail to provide control ability beyond the sparse joints of major body skeleton, leading to simple and unnatural animation.

In this work, our objective is to enhance the fine-grained control capabilities of GAN-based human avatar generation model. To achieve this, we introduce the first e**X**pressive 3D human **A**vatar **Gen**eration model (**XAGen**) that can (1) synthesize high-quality 3D human avatars with diverse realistic appearances and detailed geometries; (2) provide independent control capabilities for fine-grained attributes, including body poses, hand poses, jaw poses, shapes, and facial expressions.

XAGen is built upon recent unconditional 3D-aware generation models for static images [7, 44]. One straightforward approach to implement fully animatable avatar generation is extending 3D GAN models to condition on expressive control signals, such as SMPL-X [47]. Though conceptually simple, such a direct modification of conditioning signal cannot guarantee promising appearance quality and control ability, particularly for two crucial yet challenging regions, *i.e.*, the face and hands. This is because (1) Compared with body, face and hands contain similar or even more articulations. In addition, their scales are much smaller than arms, torso, and legs in a human body image, which hinders the gradient propagation from supervision. (2) Face and hands are entangled with the articulated human body and thus will be severely affected by large body pose deformation, leading to optimization difficulty when training solely on full-body image collections.

To address the above challenges, we decompose the learning process of body, face, and hands by adopting a multi-scale and multi-part 3D representation and rendering multiple parts independently using their respective observation viewpoints and control parameters. The rendered images are passed to multi-part discriminators, which provide multi-scale supervision during the training process.

With these careful designs, XAGen can synthesize photo-realistic 3D human avatars that can be animated effectively by manipulating the corresponding control parameters for expressions and poses, as depicted in Figure 1. We conduct extensive experiments on a variety of benchmarks [18, 68, 14, 36], demonstrating the superiority of XAGen over state-of-the-arts in terms of appearance, geometry, and controllability. Moreover, XAGen supports various downstream applications such as text-guided avatar creation and audio-driven animation, expanding its potential for practical scenarios.

Our contributions are three-fold: (1) To the best of our knowledge, XAGen is the first 3D GAN model for fully animatable human avatar generation. (2) We propose a novel framework that incorporates

multi-scale and multi-part 3D representation together with multi-part rendering technique to enhance the quality and control ability, particularly for the face and hands. (3) Experiments demonstrate XAGen surpasses state-of-the-art methods in terms of both quality and controllability, which enables various downstream applications, including text-guided avatar synthesis and audio-driven animation.

## 2 Related work

**Generative models for avatar creation.** Generative models [27, 28, 51] have demonstrated unprecedented capability for synthesizing high-resolution photo-realistic images. Building upon these generative models, follow-up works [7, 44, 55, 59, 63] have focused on extending 2D image generation to the 3D domain by incorporating neural radiance field [42] or differentiable rasterization [29]. Although enabling 3D-aware generation, these works fail to provide control ability to manipulate the synthesized portrait images. To address this limitation, recent research efforts [64, 26, 43, 69, 57, 16, 71] have explored animatable 3D avatar generation leveraging parametric models for face [32] and body [38]. These works employ inverse [31] or forward [9] skinning techniques to control the facial attributes or body poses of the generated canonical avatars [69, 71]. For human body avatars, additional challenges arise due to their articulation properties. Consequently, generative models for human avatars have explored effective 3D representation designs. Among them, ENARF [43] divides an efficient 3D representation [7] into multiple parts, with each part representing one bone. EVA3D [26] employs a similar multi-part design by developing a compositional neural radiance field. Despite enabling body control, such representation fails to generate the details of human faces or hands since these parts only occupy small regions in the human body images.

Our method differs in two aspects. First, existing works can either control face or body, whereas ours is the first 3D avatar generation model with simultaneous fine-grained control over the face, body, and hands. Second, we devise a multi-scale and multi-part 3D representation, allowing for generating human body with high fidelity even for small regions like face and hands.

**Expressive 3D human modeling.** Existing 3D human reconstruction approaches can be categorized into two main categories depending on whether explicit or implicit representations are used. Explicit representations mainly utilize the pre-defined mesh topology, such as statistical parametric models [38, 62, 45, 2] or personalized mesh registrations [19, 12], to model naked human bodies with various poses and shapes. To enhance the expressiveness, recent works have developed expressive statistical models capable of representing details beyond major human body [47, 46, 17] or introduced the surface deformation to capture fine-grained features [30, 58]. On the other hand, leveraging the remarkable advances in implicit neural representations [41, 42], another line of research has proposed to either rely purely on implicit representations [53] or combine it with statistical models [61, 48, 8] to reconstruct expressive 3D human bodies. The most recent work [15, 54] proposed to learn a single full-body avatar from multi-part portrait videos or 3D scans. In contrast, our approach focuses on developing 3D generative model for fully animatable human avatars, which is trainable on only unstructured 2D image collections.

## 3 Method

In this section, we introduce XAGen, a 3D generative model for synthesizing photo-realistic human avatars with expressive and disentangled controllability over facial expression, shape, jaw pose, body pose, and hand pose. Figure 2 depicts the pipeline of our method.

Given a random noise $\mathbf{z}$ sampled from Gaussian distribution, XAGen first synthesizes a human avatar with canonical body, face, and hand configurations. In this work, we use X-pose [34] and neutral shape, face, and hand as canonical configurations. We leverage Tri-plane [7] as the fundamental building block of 3D representation in our canonical generator. To increase the capability of 3D representation for the smaller-scale face and hands, we introduce multi-part and multi-scale designs into the canonical Tri-plane (Sec. 3.1). A mapping network first encodes $\mathbf{z}$ and the camera viewpoint of body $c_{\mathrm{b}}$ into latent code $\mathbf{w}$. The canonical generator then synthesizes three Tri-planes $\mathcal{F}_k$ conditioned on $\mathbf{w}$, where $k \in \{\mathrm{b}, \mathrm{f}, \mathrm{h}\}$ which stands for $\{\mathrm{body}, \mathrm{face}, \mathrm{hand}\}$.

Based on the generated canonical avatar, we deform it from canonical space to observation space under the guidance of control signal $p_{\mathrm{b}}$ parameterized by an expressive statistical full body model, *i.e.*, SMPL-X [47]. We adopt volumetric rendering [39] to synthesize the full body image. However, due

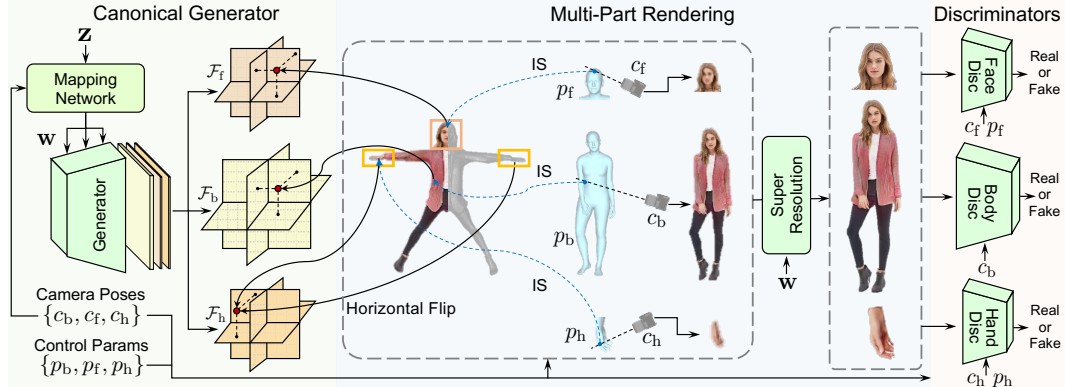

Figure 2: Pipeline of XAGen. Given a random noise $\mathbf{z}$, the canonical generator synthesizes the avatar in the format of canonical multi-part and multi-scale Tri-planes given the corresponding camera pose $c_{\mathrm{b}}$. We then deform the canonical avatar under the guidance of control parameters $p_*$ to render multi-part images using respective camera poses $c_*$ and upsample the images using a super-resolution module. Discriminators encode the output images, camera poses, and control parameters into real or fake probabilities to critique the rendered images. IS represents inverse skinning.

to the scale imbalance between the face/hands and body, rendering only the full body image cannot guarantee quality for these detailed regions. To address this issue, we propose a multi-part rendering technique (Sec. 3.2). Specifically, we employ part-aware deformation and rendering based on the control parameters ($p_{\mathrm{f}}$ and $p_{\mathrm{h}}$) and cameras ($c_{\mathrm{f}}$ and $c_{\mathrm{h}}$). Accordingly, to ensure the plausibility and controllability of the generated avatars, we develop multi-part discriminators to critique the rendered images (Sec. 3.3).

## 3.1 Multi-scale and Multi-part Representation

XAGen is designed for expressive human avatars with an emphasis on the high-quality face and hands. However, the scale imbalance between face/hands and body may hamper the fidelity of the corresponding regions. To address this issue, we propose a simple yet effective multi-scale and multi-part representation for expressive human avatar generation. Our multi-scale representation builds upon the efficient 3D representation, *i.e.*, Tri-plane [7], which stores the generated features on three orthogonal planes. Specifically, we design three Tri-planes for body, face, and hands, denoted as $\mathcal{F}_{\mathrm{b}} \in \mathbb{R}^{W_{\mathrm{b}} \times W_{\mathrm{b}} \times 3C}$, $\mathcal{F}_{\mathrm{f}} \in \mathbb{R}^{W_{\mathrm{f}} \times W_{\mathrm{f}} \times 3C}$, and $\mathcal{F}_{\mathrm{h}} \in \mathbb{R}^{W_{\mathrm{h}} \times W_{\mathrm{h}} \times 3C}$, respectively. The size of the face and hand Tri-planes is set to half of the body Tri-plane, with $W_{\mathrm{f}} = W_{\mathrm{h}} = W_{\mathrm{b}}/2$.

As depicted in Figure 2, our canonical generator first synthesizes a compact feature map $\mathcal{F} \in \mathbb{R}^{W_{\mathrm{b}} \times W_{\mathrm{b}} \times 9C/2}$, where $C$ represents the number of channels. We then separate and reshape $\mathcal{F}$ into $\mathcal{F}_k$, where $k \in \{\mathrm{b}, \mathrm{f}, \mathrm{h}\}$, representing the canonical space of the generated human avatar. Furthermore, to save computation cost, we exploit the symmetry property of hands to represent both left and right hands using one single $\mathcal{F}_{\mathrm{h}}$ through a horizontal flip operation (refer to Appendix for details).

## 3.2 Multi-part Rendering

Our method is trainable on unstructured 2D human images. Although this largely reduces the difficulty and cost to obtain data, the training is highly under-constrained due to the presence of diverse poses, faces, and clothes. To facilitate the training process and improve the appearance quality, we propose a multi-part rendering strategy. This strategy allows XAGen to learn each part based on the independent camera poses, which further enhances the geometry quality of the face and hands. Specifically, for each training image, we utilize a pretrained model [17] to estimate SMPL-X parameters $\{p_{\mathrm{b}}, p_{\mathrm{f}}, p_{\mathrm{h}}\}$ and camera poses $\{c_{\mathrm{b}}, c_{\mathrm{f}}, c_{\mathrm{h}}\}$ for body, face, and hands, respectively. In the rendering stage, we shoot rays using $\{c_{\mathrm{b}}, c_{\mathrm{f}}, c_{\mathrm{h}}\}$ and sample points $\{\mathbf{x}_{\mathbf{o}}^{\mathrm{b}}, \mathbf{x}_{\mathbf{o}}^{\mathrm{f}}, \mathbf{x}_{\mathbf{o}}^{\mathrm{h}}\}$ along the rays in the observation space. To compute the feature for each point, we employ inverse linear-blend skinning [31], which finds the transformation of each point from observation space to canonical space produced by the canonical generator. Based on the parameter $p_k$, where $k \in \{\mathrm{b}, \mathrm{f}, \mathrm{h}\}$, SMPL-X yields an expressive human body model $(\mathbf{v}, \mathbf{w})$, where $\mathbf{v} \in \mathbb{R}^{N \times 3}$ represents $N$ vertices, and $\mathbf{w} \in \mathbb{R}^{N \times J}$ represents the skinning

weights of each vertex with respect to joint $J$. For each point $\mathbf{x}_\mathbf{o}^{k,i}$, where $i = 1 \cdots M_k$ and $M_k$ is the number of sampled points, we find its nearest neighbour $\mathbf{n}$ from vertices $\mathbf{v}$. We then compute the corresponding transformation from observation space to canonical space

$$T^{k,i} = (\sum_j \mathbf{w}_j^\mathbf{n} \begin{bmatrix} R_j & t_j \\ \mathbf{0} & \mathbf{1} \end{bmatrix} \begin{bmatrix} \boldsymbol{I} & \Delta^\mathbf{n} \\ \mathbf{0} & \mathbf{1} \end{bmatrix})^{-1}, \tag{1}$$

where $j = 1 \cdots J$, $R_j$ and $t_j$ are derived from $p_k$ with Rodrigues formula [6], and $\Delta^\mathbf{n}$ represents the offset caused by pose and shape for vertex $\mathbf{n}$, which is calculated by SMPL-X. Based on this inverse transformation, we can calculate the coordinates for each point in canonical space $\mathbf{x}_\mathbf{c}^{k,i}$ as

$$\mathbf{x}_\mathbf{c}^{k,i} = T^{k,i} \mathbf{x}_\mathbf{o}^{k,i}, \tag{2}$$

where we apply homogeneous coordinates for the calculation.

For the face and hands rendering, *i.e.*, $k \in \{\mathrm{f}, \mathrm{h}\}$, we directly interpolate their corresponding Tri-plane $\mathcal{F}_\mathrm{f}$ and $\mathcal{F}_\mathrm{h}$ to compute the feature $\mathbf{f}_\mathbf{c}^{\mathrm{f},i}$ and $\mathbf{f}_\mathbf{c}^{\mathrm{h},i}$. Regarding the body rendering, we first define three bounding boxes $\mathbb{B}_\mathrm{f}, \mathbb{B}_\mathrm{lh}, \mathbb{B}_\mathrm{rh}$ for face, left and right hands in canonical body space. Then, we query canonical body points that are outside these bounding boxes from body Tri-plane $\mathcal{F}_\mathrm{b}$, while the canonical points inside these boxes from $\mathcal{F}_\mathrm{f}$ and $\mathcal{F}_\mathrm{h}$. The query process for body point $\mathbf{x}_\mathbf{c}^{\mathrm{b},i}$ is mathematically formulated as

$$\mathbf{f}_\mathbf{c}^{\mathrm{b},i} = \begin{cases} Q(\mathbf{x}_\mathbf{c}^{\mathrm{b},i}, \mathcal{F}_\mathrm{f}), & \text{if } \mathbf{x}_\mathbf{c}^{\mathrm{b},i} \in \mathbb{B}_\mathrm{f}, \\ Q(\mathbf{x}_\mathbf{c}^{\mathrm{b},i}, \mathcal{F}_\mathrm{h}), & \text{if } \mathbf{x}_\mathbf{c}^{\mathrm{b},i} \in \{\mathbb{B}_\mathrm{rh}, \mathbb{B}_\mathrm{lh}\}, \\ Q(\mathbf{x}_\mathbf{c}^{\mathrm{b},i}, \mathcal{F}_\mathrm{b}), & \text{if } \mathbf{x}_\mathbf{c}^{\mathrm{b},i} \notin \{\mathbb{B}_\mathrm{f}, \mathbb{B}_\mathrm{lh}, \mathbb{B}_\mathrm{rh}\}, \end{cases} \tag{3}$$

where $Q$ denotes querying the feature for the given point from the corresponding Tri-planes.

Once the features $\mathbf{f}_\mathbf{c}^{k,i}$ are obtained, they are encoded into color $\mathbf{c}$ and geometry $d$ via two lightweight multi-layer perceptrons (MLP), where $\mathbf{c} = \mathrm{MLP}_c(\mathbf{f}_\mathbf{c}^{k,i})$. Inspired by prior works [44, 26, 69], we employ signed distance field (SDF) as a proxy to model geometry. Additionally, following [26, 69], we also query a base SDF $d_\mathbf{c}$ in the canonical space, and predict delta SDF, such that $d = d_\mathbf{c} + \mathrm{MLP}_d(\mathbf{f}_\mathbf{c}^{k,i}, d_\mathbf{c})$. We then convert the SDF value into density $\sigma = \frac{1}{\alpha}\mathrm{Sigmoid}(\frac{-d}{\alpha})$ for volume rendering, where $\alpha$ is a learnable parameter.

To handle the body features queried from multiple Tri-planes, we apply feature composition on RGB and density using a window function [37] for smoothness transition. Specifically, if point $\mathbf{x}_{\mathbf{c},\mathbf{b}}^{k,i}$ is located in the overlapping region between the body and other parts (face, right hand, and left hand), their features are sampled from both Tri-planes and linearly blended together. More details on the feature composition can be found in the Appendix. Finally, volume rendering is applied to synthesize raw images for body, face, and hands, denoted as $\{I_\mathrm{b}^\mathrm{raw}, I_\mathrm{f}^\mathrm{raw}, I_\mathrm{h}^\mathrm{raw}\}$. These raw images are then upsampled into high-resolution images $\{I_\mathrm{b}, I_\mathrm{f}, I_\mathrm{h}\}$ by a super-resolution module.

### 3.3 Multi-part Discriminators

Based on the images synthesized by XAGen generator, we design a discriminator module to critique the generation results. To ensure both the fine-grained fidelity of appearance and geometry as well as disentangled control over the full body, including face and hands, we introduce multi-part discriminators to encode images $\{I_\mathrm{b}, I_\mathrm{f}, I_\mathrm{h}\}$ into real-fake scores for adversarial training. As depicted in Figure 2, these discriminators are conditioned on the respective camera poses to encode 3D priors, resulting in improved geometries as demonstrated in our experiments. To enhance the control ability of the face and hands, we further condition face discriminator on expression and shape parameters $[p_\mathrm{f}^\psi, p_\mathrm{f}^\beta]$, and condition hand discriminator on hand pose $p_\mathrm{h}^\theta$. We encode the camera pose and condition parameters into intermediate embeddings by two separate MLPs and pass them to the discriminators. The multi-part discriminator is formulated as

$$s_k = \mathcal{D}_k(I_k, \mathrm{MLP}_k^c(c_k) + \mathrm{MLP}_k^p(p_k')), \text{ where } p_k' = \begin{cases} \varnothing, & \text{if } k = \mathrm{b} \\ [p_\mathrm{f}^\psi, p_\mathrm{f}^\beta], & \text{if } k = \mathrm{f} \\ p_\mathrm{h}^\theta, & \text{if } k = \mathrm{h} \end{cases}. \tag{4}$$

Here $s_k$ denotes the probability of each image $I_k$ being sampled from real data, and $\mathcal{D}_k$ refers to the discriminator corresponding to the specific body part $k$. For body part, no conditioning parameters are used because we empirically find that the condition for body hinders the learning of appearance.

## 3.4 Training Losses

The non-saturating GAN loss [21] is computed for each discriminator, resulting in $L_b$, $L_f$, and $L_h$. We also regularize these discriminators using R1 regularization loss [40] $L_{R1}$. To improve the plausibility and smoothness of geometry, we compute minimal surface loss $L_{Minsurf}$, Eikonal loss $L_{Eik}$, and human prior regularization loss $L_{Prior}$ as suggested in previous works [44, 69].

Due to the occlusion in the full body images, some training samples may not contain visible faces or hands. Thus, we balance the loss terms for both generator and discriminator based on the visibility of face $\mathcal{M}_f$ and hands $\mathcal{M}_h$, which denote whether face and hands are detected or not. The overall loss term of XAGen is formulated as

$$
\begin{aligned}
L_{\mathcal{G}} &= L_b^{\mathcal{G}} + \lambda_f \mathcal{M}_f \odot L_f^{\mathcal{G}} + \lambda_h \mathcal{M}_h^{\mathcal{G}} \odot L_h + \lambda_{Minsurf} L_{Minsurf} + \lambda_{Eik} L_{Eik} + \lambda_{Prior} L_{Prior}, \\
L_{\mathcal{D}} &= L_b^{\mathcal{D}} + L_{R1}^b + \lambda_f \mathcal{M}_f \odot (L_f^{\mathcal{D}} + L_{R1}^f) + \lambda_h \mathcal{M}_h \odot (L_h^{\mathcal{D}} + L_{R1}^h),
\end{aligned}
\tag{5}
$$

where $\odot$ means instance-wise multiplication, and $\lambda_*$ are the weighting factors for each term.

## 4 Experiments

We evaluate the performance of XAGen on four datasets, *i.e.*, DeepFashion [36], MPV [68], UBC [14], and SHHQ [18]. These datasets contain diverse full body images of clothed individuals. For each image in the dataset, we process it to obtain aligned body, face and hand crops, and their corresponding camera poses and SMPL-X parameters. Please refer to Appendix for more details.

### 4.1 Comparisons

**Baselines.** We compare XAGen with four state-of-the-art 3D GAN models for animatable human image generation: ENARF [43], EVA3D [26], AvatarGen [69], and AG3D [16]. All these methods utilize 3D human priors to enable the controllability of body pose. ENARF conditions on sparse skeletons, while others condition on SMPL [38] model. Additionally, AvatarGen and AG3D incorporate an extra face discriminator to enhance face quality. We adopt the official implementations of ENARF and EVA3D, and cite results from AG3D directly. As for AvatarGen, it is reproduced and conditioned on SMPL-X to align with the setup of our model.

**Quantitative comparisons.** The fidelity of synthesized image is measured by Frechet Inception Distance (FID) [24] computed between $50K$ generated images and all the available real images in each dataset. To study the appearance quality for face and hands, we further crop face (resolution $64^2$) and hands (resolution $48^2$) regions from the generated and real images to compute $FID_f$ and $FID_h$. To evaluate pose control ability, we compute Percentage of Correct Keypoints (PCK) between $5K$ real images and images generated using the same pose condition parameters of real images under a distance threshold of 0.1. To evaluate this ability in face and hand regions, we also report $PCK_f$ and $PCK_h$. Another critical evaluation for a fully controllable generative model is the disentangled control of fine-grained attributes. Inspired by previous works [13, 64], we select one attribute from {expression, shape, jaw pose, body pose, hand pose}, and modify the selected attribute while keeping others fixed for each synthesis. We then estimate the SMPL-X parameters for $1K$ generated images using a pre-trained 3D human reconstruction model [17] and compute the Mean Square Error (MSE) for the selected attribute between the input and estimated parameters.

Table 1 summarizes the results for appearance quality and pose control ability for body, face, and hands. It demonstrates that XAGen outperforms existing methods *w.r.t.* all the evaluation metrics, indicating its superior performance in generating controllable photo-realistic human images with high-quality face and hands. Notably, XAGen shows significant improvements over the most recent method AG3D, achieving more than 20% improvement in FID and $FID_f$ on both DeepFashion and UBC datasets. Additionally, XAGen achieves state-of-the-art pose control ability, with substantial performance boost in $PCK_f$, *e.g.*, a relative improvement of 40.90% on MPV dataset against baseline.

Table 2 presents the results for the disentangled control ability of XAGen compared to the baseline methods. It is worth noting that ENARF and EVA3D are not fully controllable, but we still report all the evaluation metrics for these two methods to show the controllability lower bound. Notably, the generated images of ENARF are blurry. Thus, our pose estimator cannot estimate precise jaw poses, which leads to an outlier on UBC jaw pose. In general, XAGen demonstrates state-of-the-art

Table 1: Quantitative comparisons with baselines in terms of appearance and overall control ability, with best results in **bold**. F.Ctl. indicates whether the approach generates fully controllable human body or not. *We implement AvatarGen by conditioning it on SMPL-X.

| | | DeepFashion [36] | | | | | | MPV [14] | | | | | |
| --- | --- | --- | --- | --- | --- | --- | --- | --- | --- | --- | --- | --- | --- |
| | F.Ctl. | FID↓ | FID$_f$↓ | FID$_h$↓ | PCK↑ | PCK$_f$↑ | PCK$_h$↑ | FID↓ | FID$_f$↓ | FID$_h$↓ | PCK↑ | PCK$_f$↑ | PCK$_h$↑ |
| ENARF [43] | ✗ | 68.62 | 52.17 | 46.86 | 3.54 | 3.79 | 1.34 | 65.97 | 47.71 | 37.08 | 3.06 | 3.55 | 0.67 |
| EVA3D [26] | ✗ | 15.91 | 14.63 | 48.10 | 56.36 | 75.43 | 23.14 | 14.98 | 27.48 | 32.54 | 33.00 | 42.47 | 19.24 |
| AG3D [16] | ✗ | 10.93 | 14.79 | - | - | - | - | - | - | - | - | - | - |
| AvatarGen [69]* | ✓ | 9.53 | 13.96 | 27.68 | 60.12 | 73.38 | 46.50 | 10.06 | 13.08 | 19.75 | 38.32 | 45.26 | 30.75 |
| XAGen (Ours) | ✓ | **8.55** | **10.69** | **24.26** | **66.04** | **87.06** | **47.56** | **7.94** | **12.07** | **17.35** | **48.84** | **63.77** | **32.01** |

| | | UBC [68] | | | | | | SHHQ [18] | | | | | |
| --- | --- | --- | --- | --- | --- | --- | --- | --- | --- | --- | --- | --- | --- |
| | F.Ctl. | FID↓ | FID$_f$↓ | FID$_h$↓ | PCK↑ | PCK$_f$↑ | PCK$_h$↑ | FID↓ | FID$_f$↓ | FID$_h$↓ | PCK↑ | PCK$_f$↑ | PCK$_h$↑ |
| ENARF [43] | ✗ | 36.39 | 34.27 | 32.72 | 6.90 | 7.44 | 6.37 | 79.29 | 50.19 | 46.97 | 4.43 | 4.62 | 2.71 |
| EVA3D [26] | ✗ | 12.61 | 36.87 | 45.66 | 36.31 | 55.31 | 8.38 | 11.99 | 20.04 | 39.83 | 31.24 | 37.60 | 18.38 |
| AG3D [16] | ✗ | 11.04 | 15.83 | - | - | - | - | - | - | - | - | - | - |
| AvatarGen [69]* | ✓ | 9.75 | 13.23 | 18.09 | 65.31 | 77.09 | 55.09 | 10.52 | 12.57 | 28.21 | 59.18 | 78.71 | 36.29 |
| XAGen (Ours) | ✓ | **8.80** | **9.82** | **16.72** | **69.18** | **84.18** | **55.17** | **5.88** | **10.06** | **19.23** | **65.14** | **91.44** | **38.53** |

performance for fine-grained controls, particularly in expression, jaw, and hand pose, improving upon baseline by 38.29%, 25.93%, and 33.87% respectively on SHHQ dataset which contains diverse facial expressions and hand gestures. These results highlight the effectiveness of XAGen in enabling disentangled control over specific attributes of the generated human avatar images.

Table 2: Quantitative comparisons with baselines in terms of disentangled control ability measured by MSE. We report Jaw$\times10^{-4}$ and others $\times10^{-2}$ for simplicity, with best results in **bold**. *We implement AvatarGen by conditioning it on SMPL-X.

| | DeepFashion [36] | | | | | MPV [14] | | | | |
| --- | --- | --- | --- | --- | --- | --- | --- | --- | --- | --- |
| | Exp↓ | Shape↓ | Jaw↓ | Body↓ | Hand↓ | Exp↓ | Shape↓ | Jaw↓ | Body↓ | Hand↓ |
| ENARF [43] | 13.47 | 6.30 | 5.79 | 3.14 | 9.87 | 11.21 | 4.91 | 8.36 | 2.75 | 12.90 |
| EVA3D [26] | 6.03 | 2.87 | 5.11 | 1.78 | 3.68 | 9.97 | 4.14 | 13.83 | 1.80 | 4.65 |
| AvatarGen [69]* | 4.92 | 3.06 | 5.05 | **1.23** | 3.17 | 8.98 | **3.88** | 15.22 | 1.11 | 3.47 |
| XAGen (Ours) | **4.46** | **2.77** | **3.67** | 1.26 | **2.95** | **6.31** | **3.88** | **7.43** | **0.94** | **2.23** |

| | UBC [68] | | | | | SHHQ [18] | | | | |
| --- | --- | --- | --- | --- | --- | --- | --- | --- | --- | --- |
| | Exp↓ | Shape↓ | Jaw↓ | Body↓ | Hand↓ | Exp↓ | Shape↓ | Jaw↓ | Body↓ | Hand↓ |
| ENARF [43] | 10.70 | 6.11 | **3.62** | 1.07 | 8.19 | 14.51 | 6.43 | 8.16 | 3.27 | 9.83 |
| EVA3D [26] | 7.00 | 2.98 | 5.36 | 1.00 | 2.78 | 7.43 | 4.15 | 9.26 | 1.93 | 5.15 |
| AvatarGen [69]* | 9.59 | 4.50 | 9.34 | 1.22 | 3.01 | 9.01 | 3.99 | 8.87 | 1.52 | 4.99 |
| XAGen (Ours) | **5.35** | **2.57** | 4.76 | **0.73** | **1.63** | **5.56** | **3.66** | **6.57** | **1.24** | **3.30** |

**Qualitative comparisons.** Figure 3 provides qualitative comparisons between XAGen and baselines. From the results, we observe that ENARF struggles to produce reasonable geometry or realistic images due to the limitations of low training resolution. While EVA3D and AvatarGen achieve higher quality, they still fail to synthesize high-fidelity appearance and geometry for the face and hands. In contrast, XAGen demonstrates superior performance with detailed geometries for face and hands regions, resulting in more visually appealing human avatar images. The improvement of XAGen against baseline models is also confirmed by the perceptual user study, which is summarized in Table 3. Notably, XAGen achieves the best perceptual preference scores for both image appearance ($\geq 57.2\%$) and geometry ($\geq 48.3\%$) on all the benchmark datasets.

Figure 4 showcases qualitative results for fine-grained control ability. We first observe that ENARF fails to generate a correct arm for the given body pose. Although EVA3D demonstrates a better pose condition ability, its shape conditioning ability is limited and the generated face suffers from unrealistic scaling. On the other hand, AvatarGen shows comparable results for pose and shape control. However, when it comes to expression, jaw pose, and hand pose controls, ours significantly

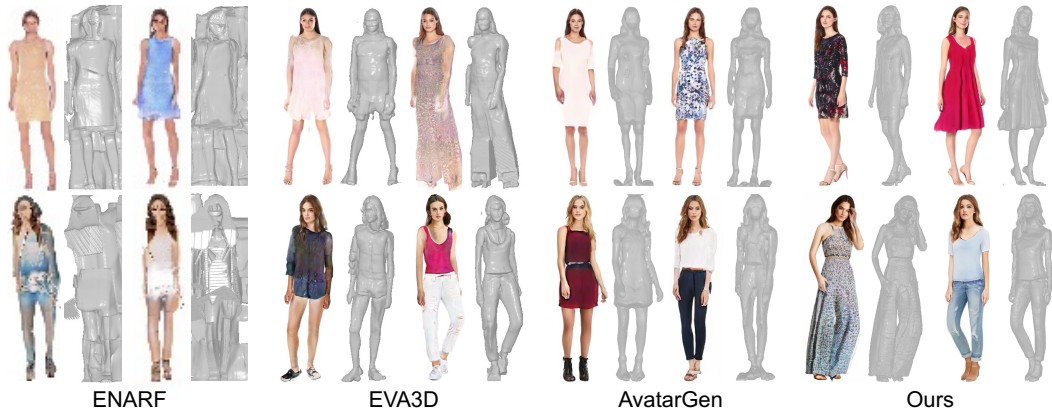

| ENARF | EVA3D | AvatarGen | Ours |

Figure 3: Comparisons against baselines in terms of appearance and 3D geometry. Our method produces photo-realistic human images with superior detailed geometries.

Table 3: We conduct a perceptual human study and report participants' preferences on images and geometries generated by our method and baselines. It is measured by preference rate (%), with best results in **bold**. *RGB* represents image, and *Geo* represents geometry. *We implement AvatarGen by conditioning it on SMPL-X.

| | DeepFashion [36] | | MPV [14] | | UBC [68] | | SHHQ [18] | |
|---|---|---|---|---|---|---|---|---|
| | *RGB*↑ | *Geo*↑ | *RGB*↑ | *Geo*↑ | *RGB*↑ | *Geo*↑ | *RGB*↑ | *Geo*↑ |
| ENARF [43] | 0.0 | 0.0 | 0.0 | 0.0 | 0.6 | 0.0 | 0.0 | 0.0 |
| EVA3D [26] | 17.3 | 35.6 | 15.0 | 17.2 | 7.8 | 34.4 | 11.3 | 15.5 |
| AvatarGen [69]* | 15.4 | 16.1 | 17.2 | 18.9 | 34.4 | 3.9 | 28.2 | 28.6 |
| XAGen (Ours) | **67.3** | **48.3** | **67.8** | **63.9** | **57.2** | **61.7** | **60.5** | **55.9** |

outperforms AvatarGen, *e.g.*, AvatarGen produces distortion in mouth region and blurred fingers while XAGen demonstrates natural faces and correct hand poses.

## 4.2 Ablation studies

To verify the design choices in our method, we conduct ablation studies on SHHQ dataset, which contains diverse appearances, *i.e.*, various human body, face, and hand poses as well as clothes.

**Representation.** XAGen adopts a multi-scale and multi-part representation to improve the quality for face and hands regions. We study the necessity of this design by removing Tri-planes for face and hands. Table 4a provides the results, indicating that using only a single full-body Tri-plane (without any specific Tri-planes for face or hands) results in a significant degradation in appearance quality. Adding either face or hand Tri-plane can alleviate this issue and all the FID metrics drop slightly. The best results are achieved when both face and hand Tri-planes are enabled, demonstrating the importance of our multi-scale and multi-part representation.

**Multi-part rendering.** In our model, we render multiple parts independently in the forward process to disentangle the learning of body, face, and hands. Table 4b demonstrates that independent rendering for face is crucial, as it significantly improves both fidelity (FID$_f$: 20.63 *vs.* 10.06) and control ability (Exp: 6.58 *vs.* 5.56, Jaw: 7.26 *vs.* 6.57) for face. Similarly, without rendering for hand, FID$_h$ increases from 18.85 to 25.94, and MSE increases from 3.28 to 4.55 (Table 4c). The effectiveness of multi-part rendering is further supported by the qualitative results shown in Figure 5. Without independent rendering, the geometry quality degrades. For example, the eyes and mouth are collapsed without face rendering, and the model also fails to synthesize geometric details for hand when hand rendering is disabled. These highlight the importance of multi-part rendering in facilitating the learning of 3D geometries for different body parts.

**Discriminators.** To study the effect of multi-part discriminators, we disable each of them during training. As shown in Table 4b, without face discriminator, the overall appearance quality deteriorates.

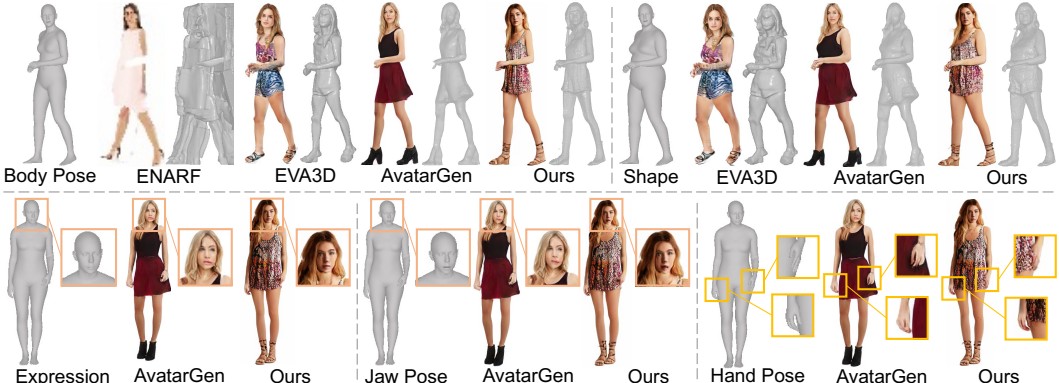

Figure 4: Qualitative comparisons in terms of disentangled control ability. Our method exhibits state-of-the-art control abilities for body pose, shape, expression, jaw pose, and hand pose.

Table 4: Ablations of our method on SHHQ dataset. We vary our representation, rendering method, and discriminators to investigate their effectiveness.

| Repr. | FID↓ | FID$_f$↓ | FID$_h$↓ |
|---|---|---|---|
| w/o both | 11.50 | 12.57 | 20.97 |
| w/ face | 11.27 | 11.95 | 20.10 |
| w/ hand | 9.64 | 11.61 | 19.92 |
| w/ both | 5.88 | 10.06 | 19.23 |

| Face | FID↓ | FID$_f$↓ | Exp↓ | Jaw↓ |
|---|---|---|---|---|
| w/o Rend | 14.53 | 20.63 | 6.58 | 7.26 |
| w/o Disc | 7.40 | 9.20 | 6.27 | 6.58 |
| w/ both | 5.88 | 10.06 | 5.56 | 6.57 |

| Hand | FID↓ | FID$_h$↓ | Hand↓ |
|---|---|---|---|
| w/o Rend | 14.28 | 26.66 | 4.51 |
| w/o Disc | 7.78 | 16.74 | 4.46 |
| w/ both | 5.88 | 19.23 | 3.33 |

(a) The effect of multi-scale and multi-part representations.

(b) The effect of face rendering and face discriminator.

(c) The effect of hand rendering and hand discriminator.

Despite the slight improvement in face appearance, there is a drop in the control ability, as evidenced by the increase in the MSE values for expression and jaw pose. A similar observation can be made for hand discriminator in Table 4c. Furthermore, the qualitative results shown in Figure 5 provide visual evidence of the impact of the face and hand discriminators on the 3D geometries. When they are removed, the geometries for face and hand collapse.

### 4.3 Applications

**Text-guided avatar synthesis.** Inspired by recent works [25, 69, 67] on text-guided avatar generation, we leverage a pretrained vision-language encoder CLIP [49] to guide the generation process using the given text prompt. The text-guided avatar generation process involves randomly sampling a latent code $z$ and a control parameter $p_b$ from the dataset, and optimizing $z$ by maximizing the CLIP similarities between the synthesized image and text prompt. As shown in Figure 6a, the generated human avatars exhibit the text-specified attributes, *i.e.*, hair and clothes adhere to the given text prompt (*e.g.*, brown hair and red T-shirt). The generated avatar can be re-targeted by novel SMPL-X parameters, allowing for additional control and customization of the synthesis.

**Audio-driven animation.** The ability of XAGen to generate fully animatable human avatars with fine-grained control (Figure 1) opens up possibilities for audio-driven animation. The 3D avatars can be driven by arbitrary SMPL-X motion sequences generated by recent works such as [66] given audio inputs. Specifically, we sample an audio stream and SMPL-X sequence from TalkSHOW [66] and use it to animate the generated avatars. As shown in Figure 6b, XAGen is able to synthesize temporally consistent video animations where the jaw poses of the avatars are synchronized with the audio stream (highlighted in red box). Additionally, the generated avatars are generalizable given novel body poses and hand gestures, allowing diverse and expressive animations.

## 5 Limitations

Although XAGen is able to synthesize photo-realistic and fully animatable human avatars, there are still areas where improvements can be made: (1) XAGen relies on pre-estimated SMPL-X parameters, the inaccurate SMPL-X may introduce potential errors into our model, which can lead to artifacts

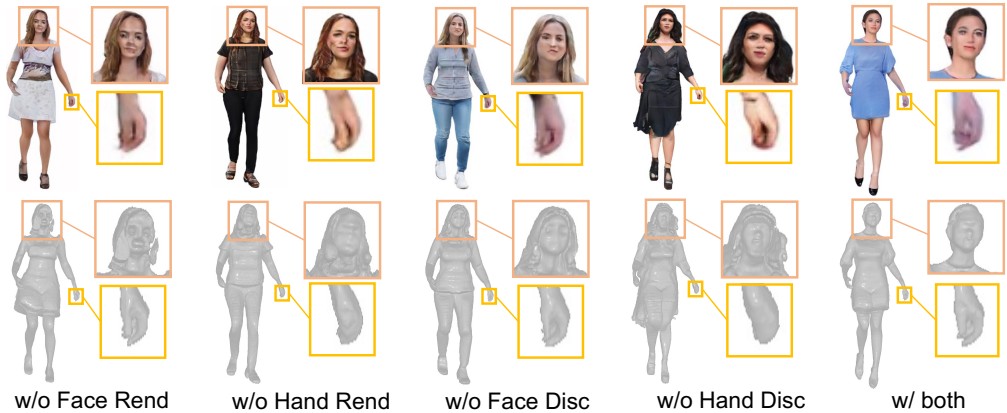

w/o Face Rend     w/o Hand Rend     w/o Face Disc     w/o Hand Disc     w/ both

Figure 5: Qualitative results for the ablations on multi-part rendering and discriminators.

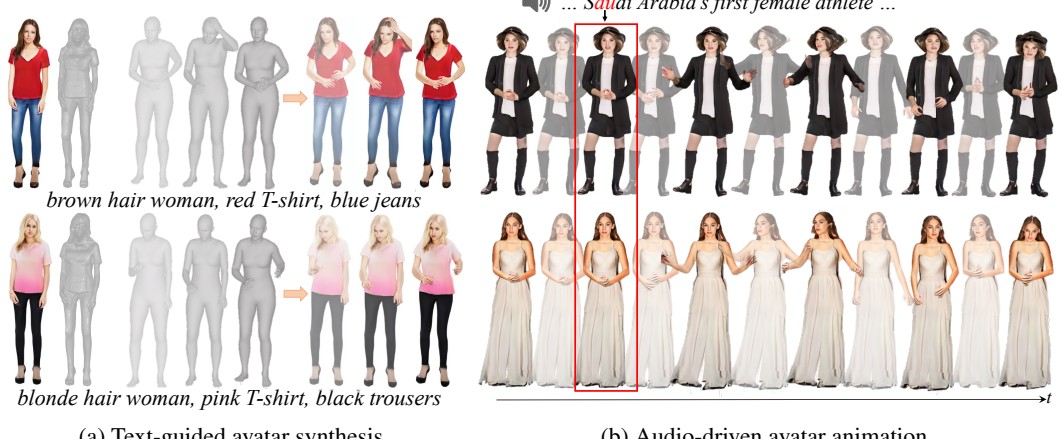

(a) Text-guided avatar synthesis.        (b) Audio-driven avatar animation.

Figure 6: Downstream applications of our method.

and degraded body images. Please refer to *Sup. Mat.* for the experimental analysis of this issue. We believe our method can benefit from a more accurate SMPL-X estimation method or corrective operations. (2) SMPL-X only represents naked body. Thus, methods built upon SMPL-X could struggle with modeling loose clothing, which is a long-standing challenge for 3D human modeling. We believe an advanced human body prior or independent clothing modeling approach is helpful to alleviate this issue. (3) Face and hand images in existing human body datasets lack diversity and sharpness, which affects the fidelity of our generation results, particularly for the novel hand poses that are out-of-distribution. A more diverse dataset with high-quality face and hand images could help tackle this problem. (4) XAGen utilizes inverse blend skinning to deform the points from canonical space to the observation space. However, this process could introduce errors, particularly when computing nearest neighbors for query points located in the connection or interaction regions. Thus, exploring more robust and accurate techniques, such as forward skinning [9], could open up new directions for future work.

## 6   Conclusion

This work introduces XAGen, a novel 3D avatar generation framework that offers expressive control over facial expression, shape, body pose, jaw pose, and hand pose. Through the use of multi-scale and multi-part representation, XAGen can model details for small-scale regions like faces and hands. By adopting multi-part rendering, XAGen disentangles the learning process and produces realistic details for appearance and geometry. With multi-part discriminators, our model is capable of synthesizing high-quality human avatars with disentangled fine-grained control ability. The capabilities of XAGen open up a range of possibilities for downstream applications, such as text-guided avatar synthesis and audio-driven animation.

## Acknowledgement

This project is supported by the National Research Foundation, Singapore under its NRFF Award NRF-NRFF13-2021-0008, and the Ministry of Education, Singapore, under the Academic Research Fund Tier 1 (FY2022).

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
