# Supplementary Material for XAGen: 3D Expressive Human Avatars Generation

## 1 Appendix

### 1.1 Feature Sampling

We provide additional details on the feature query process for face, hand, and body.

**Query face and hand features.** As described in the main paper, we directly query features for face and hand from their respective Tri-plane $\mathcal{F}_{\mathrm{f}}$ and $\mathcal{F}_{\mathrm{h}}$ to render their images. This process can be formulated as

$$\mathbf{f}_{\mathbf{c}}^{k,i} = Q(\mathbf{x}_{\mathbf{c}}^{k,i}, \mathcal{F}_k), \tag{1}$$

where $k \in \{\mathrm{f}, \mathrm{h}\}$. Here, $Q$ denotes querying the feature from Tri-plane. Specifically, this process involves grid sampling (interpolation) operation (represented as $\mathrm{Inter}$) for each query point $\mathbf{x}_{\mathbf{c}}^{k,i}$ in three orthogonal feature planes $\{\mathcal{F}_k^X, \mathcal{F}_k^Y, \mathcal{F}_k^Z\}$. Prior to the query, we normalize the coordinates of the points using the bounding boxes $\mathbb{B}_k$ defined in the canonical space for the face and hands. This normalization is performed as

$$\hat{\mathbf{x}}_{\mathbf{c}}^{k,i} = \frac{2\mathbf{x}_{\mathbf{c}}^{k,i} - (\mathbb{B}_k^{min} + \mathbb{B}_k^{max})}{\mathbb{B}_k^{max} - \mathbb{B}_k^{min}}, \tag{2}$$

Here, we use $k \in \{\mathrm{f}, \mathrm{rh}\}$, indicating that only the right hand image is rendered in multi-part rendering process. Consequently, the query process can be mathematically expressed as

$$Q(\mathbf{x}_{\mathbf{c}}^{k,i}, \mathcal{F}_k) = \sum_{t \in \{X,Y,Z\}} \mathrm{Inter}(\hat{\mathbf{x}}_{\mathbf{c}}^{k,i}, \mathcal{F}_k^t), \tag{3}$$

**Query body features.** We apply a similar process to normalize the coordinates of body points and sample them from their respective Tri-planes based on which part they belong to. In addition, during body image rendering, it is necessary to render both the left and right hands. To reduce computational cost, we exploit the symmetry between the left and right hands. Hence, we utilize a single hand Tri-plane, denoted as $\mathcal{F}_{\mathrm{h}}$ to model both hands simultaneously. Here, we explicitly define the bounding box $\mathbb{B}_{\mathrm{lh}}$ for left hand. For the body point falling within $\mathbb{B}_{\mathrm{lh}}$, we query their features by flipping the $x$-axis. Suppose a normalized left hand point is $\hat{\mathbf{x}}_{\mathbf{c}}^{\mathrm{lh},i} = [x, y, z]^T$, the flipped coordinate would be $\hat{\mathbf{x}}'^{\mathrm{lh},i}_{\mathbf{c}} = [-x, y, z]^T$. The query process for the left hand can be expressed as

$$Q(\mathbf{x}_{\mathbf{c}}^{\mathrm{lh},i}, \mathcal{F}_k) = \sum_{t \in \{X,Y,Z\}} \mathrm{Inter}(\hat{\mathbf{x}}'^{\mathrm{lh},i}_{\mathbf{c}}, \mathcal{F}_{\mathrm{h}}^t), \tag{4}$$

**Composition of body features.** As the body features are sampled from multi-part representations for the face and hand, there may be noticeable transitions in the overlapping regions between the body and other parts. To address this issue and enhance the smoothness of the transitions, we utilize a window function [12] to composite the features. As depicted in Figure 1, in addition to the bounding boxes for face and hands $\mathbb{B}_{\mathrm{f}}, \mathbb{B}_{\mathrm{lh}}, \mathbb{B}_{\mathrm{rh}}$, we further define three bounding boxes $\mathbb{B}_{\mathrm{f}}^{\mathrm{b}}, \mathbb{B}_{\mathrm{lh}}^{\mathrm{b}}, \mathbb{B}_{\mathrm{rh}}^{\mathrm{b}}$ in the canonical space for the transition regions. For each point located in the overlapping regions $\{\mathbb{B}_{\mathrm{f}} \cap \mathbb{B}_{\mathrm{f}}^{\mathrm{b}}, \mathbb{B}_{\mathrm{rh}} \cap \mathbb{B}_{\mathrm{rh}}^{\mathrm{b}}, \mathbb{B}_{\mathrm{lh}} \cap \mathbb{B}_{\mathrm{lh}}^{\mathrm{b}}\}$ (highlight in green), we normalize their coordinates based on the

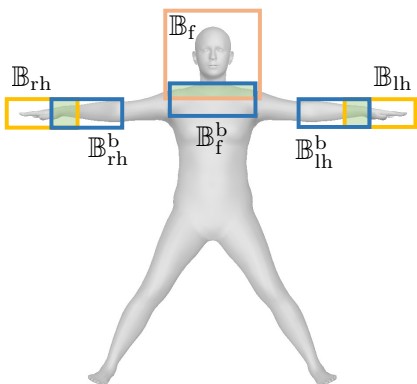

Figure 1: We define the bounding boxes ($\mathbb{B}_{\mathrm{f}}, \mathbb{B}_{\mathrm{lh}}, \mathbb{B}_{\mathrm{rh}}$)in the canonical space. Overlapping regions ($\{\mathbb{B}_{\mathrm{f}} \cap \mathbb{B}_{\mathrm{f}}^{\mathrm{b}}, \mathbb{B}_{\mathrm{rh}} \cap \mathbb{B}_{\mathrm{rh}}^{\mathrm{b}}, \mathbb{B}_{\mathrm{lh}} \cap \mathbb{B}_{\mathrm{lh}}^{\mathrm{b}}\}$) are highlighted in green.

bounding box of each part ($\hat{\mathbf{x}}'^{k,i}_{\mathbf{c}} = [x, y, z]^T$) and sample their features twice. The first sampling is performed from the body Tri-plane $\mathcal{F}_{\mathrm{b}}$, and the second sampling is performed from the corresponding part Tri-plane $\mathcal{F}_k$. Subsequently, we encode the features into color $\mathbf{c}$ and geometry $d$. The process of feature composition can be formulated as follows

$$\{\mathbf{c}, d\} = \frac{1}{\sum \omega} \sum \omega \{\mathbf{c}^k, d^k\}, \text{ where } \omega = \exp(-m(x^n + y^n + z^n)), k \in \{\mathrm{f}, \mathrm{rh}, \mathrm{lh}\}. \quad (5)$$

Here, $m$ and $n$ are empirically chosen parameters, with $m = 2$, $n = 6$ in this work.

## 1.2 Loss Terms

**Minimal surface loss.** Inspired by previous studies [23, 15, 9], we utilize the minimal surface loss to discourage the presence of spurious and invisible surfaces within the generated scene. This loss term guides the generator to create a human avatar with minimal volume of zero-crossings. Specifically, the SDF values that are in proximity to zero will be penalized by the loss term, which encourages a smoother and more coherent surface. This process is formulated as

$$L_{\mathrm{Minsurf}} = \sum_{k,i} \exp(-100|d^{k,i}|), \quad (6)$$

**Eikonal loss.** The Eikonal term is derived from the Eikonal equation, which ensures that the SDF defines a smooth boundary by enforcing its differentiability everywhere, specifically $||\nabla d^{k,i}|| = 1$. Consequently, the Eikonal loss is defined as

$$L_{\mathrm{Eik}} = \sum_{k,i} (||\nabla d^{k,i}|| - 1). \quad (7)$$

**SMPL-X prior loss.** Despite representing a naked human body, the coarse geometric information encoded in the SMPL-X parametric model can still provide valuable guidance for training our avatar generator. Hence, we incorporate a regularization term to align the predicted geometry value $d$ with the corresponding SDF value $d_{\mathbf{c}}$ queried from the canonical SPML-X space. This loss term is expressed as

$$L_{\mathrm{Prior}} = \frac{1}{|\mathcal{R}|} \sum_{\mathbf{x}_{\mathbf{c}}^{k,i} \in \mathcal{R}} w||(d^{k,i} - d_{\mathbf{c}}^{k,i})||, \text{ where } w = \exp\left(\frac{-(d_{\mathbf{c}}^{k,i})^2}{\kappa}\right), \quad (8)$$

where $\mathcal{R}$ represents the set of sampled rays.

## 1.3 Inference

XAGen synthesizes the canonical avatar and renders body, face, and hand images using their respective cameras during training. However, in the inference stage, we exclude the part-aware rendering for

face and hands. This means that we only render images for the full bodies as XAGen focuses on human avatar generation. Even though only body image rendering is enabled during inference, we continue to utilize the same multi-scale and multi-part 3D representation. This canonical avatar representation still preserves the fine details for the face and hands and provides control ability for these regions.

## 1.4 Implementation Details

We implement our model using PyTorch [16], and optimized it using the Adam optimizer. The learning rate for the generator is set to $2.5 \times 10^{-3}$, while for the discriminator it is set to $2 \times 10^{-3}$. During the training stage, we employ a volume rendering resolution of $224 \times 112$ for the full body, and $28^2$ for the face and hands. Additionally, the super-resolution module upsamples body image into $512 \times 256$, face image into $256^2$, and hand image into $256^2$. For training, we utilize a depth resolution of 40, with 20 for hierarchical sampling, while we use a depth resolution of 64 in inference stage. The weighting factors for the loss terms are set as follows: $\lambda_f = 0.25$, $\lambda_h = 0.75$, $\lambda_{\text{Minsurf}} = 5 \times 10^{-3}$, $\lambda_{\text{Eik}} = 1 \times 10^{-3}$, and $\lambda_{\text{Prior}} = 1.0$. We apply R1 regularization with a R1 gamma value of 10. Our model is trained on 8 Nvidia V100 GPUs for 106 hours with a batch size of 16.

## 1.5 Quantitative Results for Geometry

In order to quantitatively evaluate the geometry quality of XAGen, we employ depth metrics as suggested in a recent work [23]. We generate $1K$ images and adopt a pretrained human digitalization model [19] to estimate the pseudo ground truth depth map for the synthesized images. We then calculate the Mean Square Error (MSE) between the pseudo labels and the depth maps rendered by volume rendering to assess the geometric quality. We compare XAGen with the strongest baseline, AvatarGen, on four benchmark datasets. We report Depth following AvatarGen (at a resolution of $128^2$) and we also report Depth256 which is calculated at a higher resolution of $256^2$ to evaluate the details of the geometry.

Table 1: Quantitative comparisons of geometric quality between XAGen and AvatarGen on four benchmark datasets, with best results in **bold**. $^*$We implement AvatarGen by conditioning it on SMPL-X.

|  | DeepFashion [11] | | MPV [5] | | UBC [22] | | SHHQ [8] | |
|---|---|---|---|---|---|---|---|---|
|  | Depth↓ | Depth256↓ | Depth↓ | Depth256↓ | Depth↓ | Depth256↓ | Depth↓ | Depth256↓ |
| AvatarGen [23]$^*$ | .794 | .950 | .688 | .867 | .806 | .958 | .826 | .932 |
| XAGen (Ours) | **.474** | **.597** | **.527** | **.651** | **.482** | **.613** | **.438** | **.560** |

Table 1 summarizes the results for Depth metrics. XAGen consistently outperforms the baseline method in terms of depth consistency for both Depth and Depth256 metrics across all datasets. It demonstrates that our method achieves a superior geometric quality, *e.g.*, a relative improvement of 47.0% for Depth and 39.9% for Depth256 on SHHQ dataset.

## 1.6 Additional Qualitative Results

**Qualitative comparisons.** We visualize more results in Figure 2 for qualitative comparisons on MPV and SHHQ datasets. XAGen outperforms baseline methods in terms of both visual quality and geometric details, which is also confirmed by our perceptual human study.

**Disentanglement between identity and condition.** Figure 3 shows that XAGen can generate avatars with different identities under the same jaw and pose conditions. We further apply random walk between two latent codes to demonstrate this disentangled generation ability.

**Animation using AMASS.** Figure 4 shows more animation results driven by the realistic human motion sequences sampled from AMASS [13]. It demonstrates that our generated avatars can be animated by the motion capture results, which could open up possibilities for downstream applications.

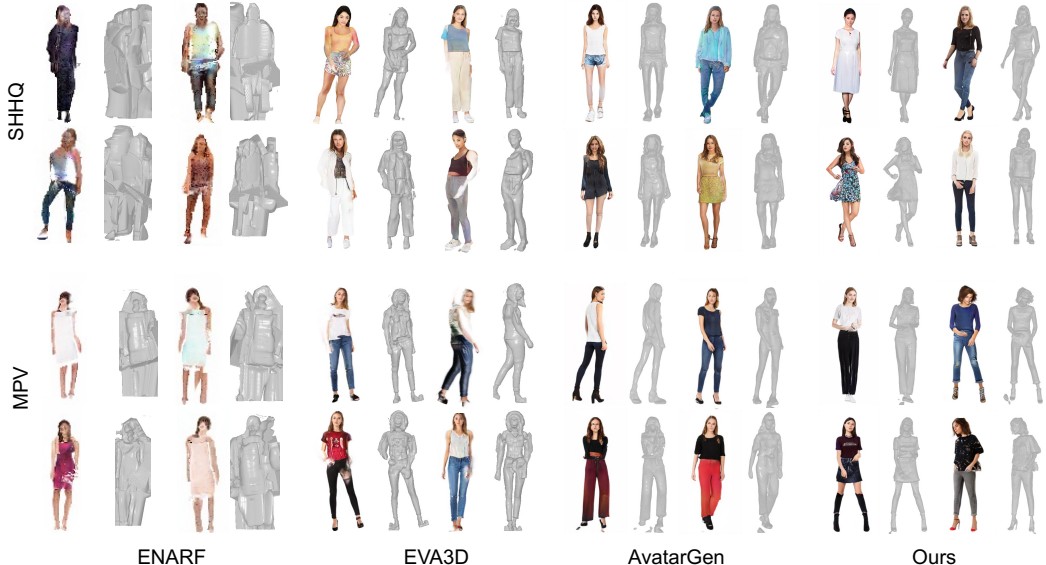

Figure 2: Qualitative comparisons between XAGen and baselines on SHHQ and MPV datasets. Best view in $2\times$ zoom.

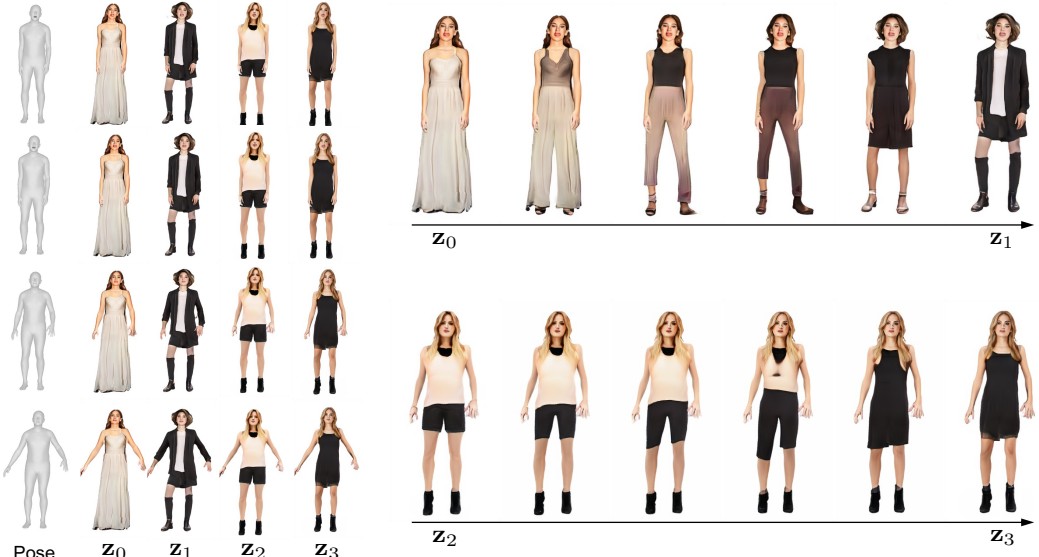

Figure 3: Generation results using various latent codes under different jaw and body poses. We do random walk in latent space to demonstrate the disentanglement between identity and condition.

## 1.7 Additional Ablation Studies

**SMPL-X errors.** XAGen generates avatars based on the pre-estimated SMPL-X parameters. Therefore, the SMPL-X estimation error could jeopardize the training process. To study its effects, we manually add random noises into the existing SMPL-X estimation results during training. The results reported in Table 2a show that noisy SMPL-X decreases the model's performance in terms of all the evaluation metrics. It mainly affects the control ability and full body image fidelity. Therefore, a more precise SMPL-X estimation method is necessary for improving control ability and full body image quality.

**Discriminator conditions.** (1) *Facial Expression:* The results in Table 2b show that although adding expression condition slightly decreases the face image fidelity, it will improve the control ability for facial expression. Thus, we condition the face discriminator on expression parameters. (2) *Body Pose:*

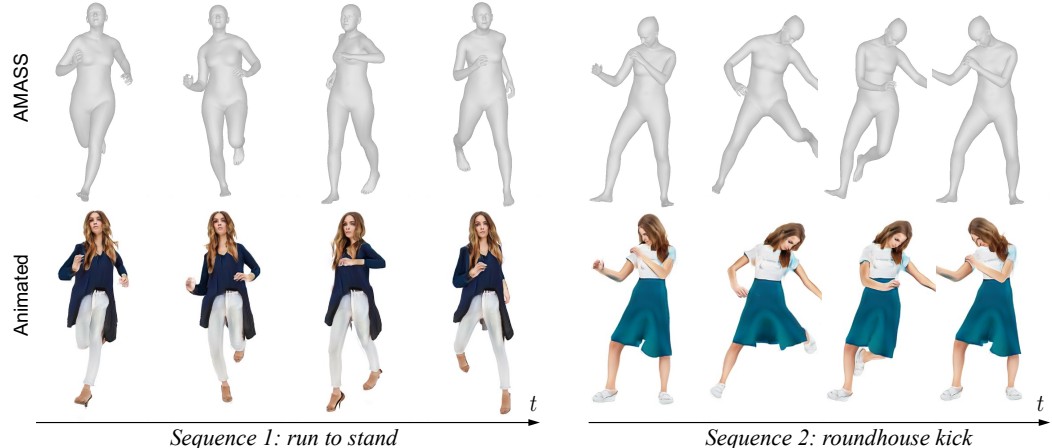

Figure 4: Animation results using motion sequences sampled from AMASS [13].

Table 2: Additional ablations of our method on SHHQ dataset. We study the effects of SMPL-X errors, discriminator conditions, shared generator, and shared hand Tri-plane.

| SMPL-X | FID↓ | FID$_f$↓ | FID$_h$↓ | PCK↑ | PCK$_f$↑ | PCK$_h$↑ | Exp↓ | Shape↓ | Jaw↓ | Body↓ | Hand↓ |
|---|---|---|---|---|---|---|---|---|---|---|---|
| noisy | 8.61 | 10.48 | 19.55 | 62.17 | 90.48 | 31.14 | 6.78 | 3.88 | 7.45 | 1.51 | 3.75 |
| clean | 5.88 | 10.06 | 19.23 | 65.14 | 91.44 | 38.53 | 5.56 | 3.66 | 6.57 | 1.24 | 3.30 |

(a) The effect of SMPL-X estimation errors.

| Expression | FID$_f$↓ | PCK$_f$↑ | Exp↓ |
|---|---|---|---|
| w/o | 9.57 | 91.43 | 5.86 |
| w/ | 10.06 | 91.44 | 5.56 |

(b) The effect of facial expression condition in face discriminator.

| Body Pose | FID↓ | PCK↑ | Body↓ |
|---|---|---|---|
| w/ | 14.23 | 66.69 | 1.00 |
| w/o | 5.88 | 65.14 | 1.24 |

(c) The effect of body pose condition in body discriminator.

| Hand Pose | FID$_h$↓ | PCK$_h$↑ | Hand↓ |
|---|---|---|---|
| w/o | 19.57 | 40.07 | 3.59 |
| w/ | 19.23 | 38.53 | 3.30 |

(d) The effect of hand pose condition in hand discriminator.

| Generator | FID↓ | FID$_f$↓ | FID$_h$↓ |
|---|---|---|---|
| separated | 7.65 | 12.20 | 21.03 |
| shared | 5.88 | 10.06 | 19.23 |

(e) The effect of sharing generator branches for body, face, and hand.

| Hand Triplanes | FID↓ | FID$_f$↓ | FID$_h$↓ | PCK$_h$↑ | Hand↓ |
|---|---|---|---|---|---|
| double | 8.32 | 10.12 | 20.53 | 39.64 | 3.27 |
| single | 5.88 | 10.06 | 19.23 | 38.53 | 3.30 |

(f) The effect of sharing Tri-planes for two hands.

It can be observed in Table 2c that conditioning on body pose will significantly affect image quality. Although it can improve the control ability for body pose, the quality decrease is too large. Thus, we do not condition the discriminator on body pose. We think the reason could be the body pose parameter itself. In SMPL-X, all the poses are encoded by relative angle-axis representations. It is too difficult for the discriminator to decode such abstract geometric transformations. (3) *Hand Pose:* Table 2d summarizes the effect of hand pose condition in discriminator. We can see that although the PCK$_h$ slightly drops by 3.8%, both the visual quality and the MSE for hand control are improved. Thus, we choose to condition the hand discriminator on hand pose.

**Shared generator.** In Table 2e, we study the effects of using separated generator branches for body, face, and hand. It shows that using separated generators cannot improve generation quality. We think the reason would be the redundancy in generators. The redundancy may increase the computation cost and hinder the optimization of the generator.

**Shared hand Tri-plane.** XAGen utilizes one shared hand Tri-planes for two hands. Table 2f shows that using two separated hand Tri-planes can improve the control ability of hand, but image fidelity decreases. We think the reason is that our generator learns to generate an additional hand Tri-plane, which is difficult to optimize. Although independent hand Tri-planes can help model

different accessories, accessories usually have small scales. Compared with the full body image, this improvement is less important and will not affect the overall image fidelity too much.

## 1.8 Dataset Preprocessing Pipeline

The datasets used in our experiments include DeepFashion [11], MPV [5], UBC [22], and SHHQ [8]. We follow a specific pipeline to process these datasets, as outlined below:

1. *Foreground Mask Estimation*: We employ a segmentation model [3] to estimate the foreground human body mask for each image. The background is then removed by padding the masked regions with white color.

2. *Keypoint Detection*: A keypoint estimation model [6] is used to detect full body keypoints. Based on these keypoints, we crop and align the body, face, and hand images following the method described in [10].

3. *SMPL-X Parameter Estimation*: We estimate the SMPL-X [17] parameters for the body, face, and hands using a pretrained 3D human reconstruction model [7].

4. *Image Cropping and Resizing*: The images are cropped once again to move the center joint to the center of the field-of-view. The body image is resized to $512 \times 256$, while the face and hand images are resized to $256^2$.

5. *Camera Parameters*: We utilize the global orientation and translation from the estimated SMPL-X parameters as the camera extrinsics for each image crop. Additionally, we assume camera focal lengths of 2560, 6400, and 8000 for body, face, and hand camera intrinsics, respectively.

After processing the datasets with the above pipeline, we obtain the following numbers of training images: $39K$ for SHHQ, $12K$ for DeepFashion, $16K$ for MPV, and $33K$ for UBC. To augment the training data, we follow the approach in [1] and horizontally flip the images and SMPL-X parameters.

## 1.9 Details for Evaluation Metrics

**Frechet inception distance.** XAGen generates $512 \times 256$ images for full body, which is different from the square images with a resolution of $512^2$ used in most previous works. We therefore follow EVA3D [9] to pad the images to $512^2$ to ensure a fair comparison. To compute $FID_f$ and $FID_h$, we leverage the estimated ground truth keypoints derived from the SMPL-X to crop the face and hands regions. This ensures that only the relevant regions are considered in the FID calculation, allowing for a more accurate evaluation of the generated face and hand images.

**Percentage of correct keypoints.** We adopt a pretrained 2D whole body pose estimation model implemented by [2] to estimate 136 keypoints defined by Halpe benchmark [6]. While previous works [23, 9, 14] typically use a threshold of 0.5 times the head length to compute PCK, we found that this threshold is too large for the fine-grained keypoints located on the face and hands. Therefore, we use a smaller threshold of 0.1 times the head length to measure the detailed pose control ability. This smaller threshold allows us to assess the accuracy of keypoint localization for these specific regions, providing a more precise evaluation of the pose control capabilities.

**Mean square error for disentangled control abilities.** We follow the evaluation method suggested in prior works [20, 4] to measure the disentangled control abilities. For an expressive human avatar, we evaluate the attributes in {shape, expression, jaw pose, body pose, hand pose}. When testing each attribute, we randomly sample one novel control parameter from the dataset and replace the parameters for this attribute with the sampled parameter, while the other attributes remain unchanged. Then, we use the 3D human reconstruction model [7] to estimated SMPL-X for the synthesized images. In [20, 4], the control ability is measured by the covariance of the estimated parameters for the selected attribute against the unchanged attributes. However, we find this measurement inaccurate for human body because some works, such as ENARF, cannot produce a clear human body image, leading to inaccurate SMPL-X estimation which causes a large and unstable covariance. Therefore, instead of calculating the covariance, we measure the Mean Square Error (MSE) for each attribute, which can further reflect the fine-grained manipulation ability. In addition, because ENARF and EVA3D are not fully animatable models, we only evaluate body pose for ENARF, and evaluate shape and body pose for EVA3D.

**Mean square error for 3D geometry.** To evaluate the geometry quality quantitatively, we follow the prior work [23] to estimate pseudo ground truth depth map for the generated image. and measure the consistency between pseudo ground truth and the rendered depth map. To measure the depth consistency on both coarse and fine levels, we report Depth for $128^2$ and Depth256 for $256^2$. To ensure fair comparisons with previous works, we pad the depth map to a square image format.

**User study.** To compare the generation performance, we conduct a human study for both appearance and geometry. We generate images and the corresponding geometries for four benchmark datasets. We generate 320 samples and recruit 20 participants to measure human preference scores.

## 1.10 Details for Baselines

**ENARF.** We use the official online implementation[1]. This includes processing the datasets and training the baseline models according to the default settings provided by the authors.

**EVA3D.** For MPV, SHHQ, and UBC datasets, we follow the instructions provided by the authors to process the datasets and estimate SMPL parameters. As for DeepFashion, we use the processed dataset released by the authors. We directly use the checkpoints released by the authors to compute the evaluation metrics for DeepFashion, SHHQ, and UBC. While for MPV, we train EVA3D on it for $430K$ iterations using the official implementation[2].

**AvatarGen.** We reproduce AvatarGen model and modified it to condition on SMPL-X parameters, enabling full body controllability. We follow the descriptions in their paper [23] while change their human prior from the observation space to the canonical space to ensure fair comparisons. This also reduces its heavy computation cost for querying SDF values in each iteration.

## 1.11 Details for Applications

**Text-guided avatar synthesis.** For text-guided avatar synthesis, the process involves randomly sampling a latent code $\mathbf{z}$ and a SMPL-X parameter from the training dataset. The latent code is optimized for 100 iterations. While the CLIP [18] loss is used to guide the synthesis process, it may lack supervision for detailed regions such as the face. To address this, a perceptual [24] loss based on the face region is computed to enforce consistency with the image of the original latent code $\mathbf{z}$. To retarget the synthesized avatar, we randomly sample SMPL-X parameters from an unseen sequence provided by [21].

**Audio-driven animation.** The generated avatars are driven by the motion sequences sampled from TalkSHOW [21] dataset. However, there is a dimension mismatch between the expression and shape parameters of TalkSHOW and the SMPL-X model we adopt. TalkSHOW uses a dimension of 100 for expression and a dimension of 300 for shape, while the default SMPL-X model in our approach uses 50 parameters for expression and 200 parameters for shape. To address this domain gap, we export the body pose, jaw pose, and hand pose from TalkSHOW, while keep the shape and expression parameters unchanged for one avatar. The complete video results for audio-driven animation can be found on our Project Page.

## 2 Broader Impact

Our model could be used to generate fake individuals or manipulate images for malicious purposes, such as misinformation, harassment, or fraud. These harmful applications could possibly pose a societal threat.

The datasets used to train our model may have inherent biases, such as unbalanced demographic distributions. Therefore, our model may reflect such biases presented in the datasets. It is essential to be mindful of these biases and consider fairness issues when deploying the model.

To protect our model from misuse and implement safeguards, we will resort to license agreements for model download and usage, which adds restrictions on the access and applications.

---

[1]https://github.com/nogu-atsu/ENARF-GAN
[2]https://github.com/hongfz16/EVA3D

# 3 Reproducibility

In this supplementary material, we provide comprehensive information to ensure the reproducibility of our work. We introduce the implementation details (Section 1.4), dataset pre-processing pipeline (Section 1.8), and details for evaluation metrics (Section 1.9). In addition, we provide code for data pre-processing, training, and evaluation.