# OpenReview forum: "XAGen: 3D Expressive Human Avatars Generation"
_NeurIPS.cc/2023/Conference — NeurIPS 2023 poster_

### Official Review · Reviewer_7EQQ · 2023-06-30

**Soundness:** 3 good
**Presentation:** 3 good
**Contribution:** 2 fair
**Rating:** 7
**Confidence:** 5

**Summary:**

The paper proposes a generative model for 3D expressive human avatars. Based on the backbone of the recent works, tri-plane 3D feature representation, volumetric rendering, transformation from pose space to canonical space, parametric body models and GAN, the authors propose to improve the expressive of the model by representing the hand and face with extra tri-planes and supervising them with extra adversarial loss. The result achieves state of the art performance according to various evaluation metrics on various dataset. Moreover, one can animate the hand and jaw of the generated avatar, surpassing most SOTA approaches. Two practical applications of such a generative model are demonstrated in the paper.

**Strengths:**

1. The paper presents a state of the art generative model for 3D clothed human avatar with expressive control.

2. The paper is solid, well written and easy to follow.

3. The evaluation is extensive and the results are convincing.

4. Two practical applications are demonstrated.

**Weaknesses:**

1. Lack of technical novelty, multi-tri-plane and part focused supervision are quite weak when claimed as novelty. But I'm relatively ok with this since the performance and the control of the model surpass the SOTA. Although in my opinion, this paper fits much more for SIGGRAPH/Asia, where technical novelty is less expected compared to NeurIPS, I believe the overall quality of the submission clearly reaches the acceptance standard for NeurIPS.

2. While I appreciate the effort on making consistent and good format of references (I notice the authors use consistent form of names of the same conference across the reference, which is rare nowadays in most of the submissions), there is still room to improve the reference, such as
a. Capitalization of words. For example: 3d -> 3D, Icon: implicit... -> ICON: Implicit..., Avartargen: a 3d... -> AvartarGen: A 3D...,
b. the URL in the references is not necessary.
c. [65] In ECCVw, 2023 -> In ECCV Workshops, 2022

3. In the supplementary video,
a. the voice over contains environment noise and echoes. It would be great to remove them.
b. the bottom left animation has a white background, occluding the slide contents. It could be better if those animations were rendered with transparent background.
c. the demonstration of controlling jaw motion and hand poses are a bit long and not eye catching. It would be more appealing if they can be shortened a bit and shown with a close-up / zoom-in view.

4. On the project page, the RGB video and the geometry visualization are asynchronized regardless of refreshing the page.

5. Suggestion. Please consider incorporating some concurrent work into related work for the updated version to make the paper more inclusive, for example: Chupa (https://snuvclab.github.io/chupa/), a generative model for 3D clothed human geometry; and SCARF (https://yfeng95.github.io/scarf/), creating an expressive avatar from monocular video.

**Questions:**

The submission is in a good state, method clearly stated, experiment is extensive and convincing. I don't have further questions to the authors regarding reviewing this paper.

**Limitations:**

Limitations and societal impact are discussed.

---

> ### Author Rebuttal · Authors · 2023-08-09
>
> We thank the reviewer for the positive feedback and recognition that 1) our method achieves state-of-the-art results; 2) our work is solid and its overall quality is good; 3) our experiments are extensive and convincing. We respond to each of your comments one by one in what follows.
>
> > **Weakness 1**
>
> We would like to thank the reviewer for the recognition that our overall quality is good for this conference. While we acknowledge the individual components used in our work have been explored in previous studies, combining and adapting them for generative and expressive 3D human modeling is a challenging and non-trivial task. Our sophisticated design not only generates high-quality 3D fully animatable avatars but also achieves clear improvements in face and hands generation/animation. Moreover, our method facilitates various downstream applications, like text-guided synthesis and video/audio-driven animation. Thus, we believe integrating all the important modules coherently and demonstrating their effects through a comprehensive ablation study constitutes a valuable contribution, as also highlighted by Reviewer 3QMX.
>
> > **Weakness 2**
>
> Thanks for the recognition and suggestions, we also appreciate your careful observation. We have modified the reference section accordingly and will update the final version.
>
> > **Weakness 3**
>
> Thanks for the useful suggestions, we will update the supplementary video accordingly in the final version.
>
> > **Weakness 4**
>
> We think this could be an issue with the browser. It is recommended to wait until all the samples are fully downloaded and then refresh the page. Also, we will further compress the gif files and update the webpage later to enhance the experience.
>
> > **Weakness 5**
>
> Thanks for the suggestion. We have already added the above concurrent works into the related work. We will update our final version accordingly.

---

### Official Review · Reviewer_GuK3 · 2023-07-02

**Soundness:** 4 excellent
**Presentation:** 4 excellent
**Contribution:** 2 fair
**Rating:** 5
**Confidence:** 5

**Summary:**

The paper proposes a method for the generation of high quality, articulable 3D avatars of humans. The method proposed builds on top of the 3D GAN framework which has been used to learn to generate 3D articulable human bodies from collections of 2D images of humans. Additionally, the method adopts the proposed high-level methodology of generating bodies in a canonical pose and then using an explicit deformation guided by a parametric body model in order to render images in a desired pose. The novel contribution of the method is unifying this generation architecture not only for bodies, but for faces, bodies, and hands, and demonstrating that generating, articulating, and discriminating these body parts individually results in a higher quality result than handling them all simultaneously. The paper describes the architecture which accomplishes this, and demonstrates that this results in a state-of-the-art generation quality for 3D human avatars. The contributions of modeling and applying losses to these body parts separately are ablated, demonstrating that they are responsible for improvement in the quality of generated avatars.

AFTER REBUTTAL:

I have read the authors' rebuttal. I believe the additional comparisons provided for the hand and jaw quality address my weakness there. As I mentioned, I still have some questions about the novelty of each of the components, but I think the method performs state-of-the-art overall and combining all of these components into a working method is important for the community. Thus, I am leaning positively.

**Strengths:**

In my opinion, the main strengths of the paper are that:
1. The paper is well written and is clear to read and follow, and presentation is extremely polished. As a result, I feel like the method will be impactful and those working in the field of generative 3D articulable avatars will want to build on top of it.
2. The proposed method demonstrates state-of-the-art results. The evaluations are extremely convincing both qualitatively and quantitatively, and compared to the baselines EVA3D and AvatarGen which are currently generating the state-of-the-art, I believe the generated results are significantly better. This is extremely important in pushing towards photorealistic generative 3D human avatars.
3. The method is ablated well. The main contributions: separately modeling faces and hands, and using a separate discriminator for each of them, are both ablated in a very clear way showing how they improve the performance of the method. This is extremely important to understand that the contributions proposed are actually responsible for the increase in quality, rather than just hyperparameter tuning.

**Weaknesses:**

In my opinion, there are two weaknesses of the paper.
1. The evaluation accuracy of the facial/hand articulation is not entirely convincing.
    - While Table 2 shows that there are improvements in “Jaw” and “Hand” when a facial and hand pose estimator is applied to the generated results, it is only compared to the modified AvatarGen with SMPL-X (which it outperforms). In order to actually ground these numbers with a baseline, it would be insightful to apply these pose estimators to the results of EVA3D and ENARF. I understand these methods don’t model hand or facial poses, but it would be an important comparison to understand how much improvement is gotten from even modeling these at all.
    - Quantitative comparisons don’t ever show multiple different identities with modified facial or hand poses (for example, in the supplementary video). Because of this, I don’t know if the examples provided (for example, mouth open), are cherry-picked results or consistent. It would be immensely helpful to show a result where the facial/hand pose is held constant in a deformed state (for example, mouth open), and then the identity latent space is walked through.
2. The amount of method novelty is relatively limited. While the results are very nice and this is an important contribution, the method is based off of the same framework as ENARF/GNARF/EVA3D/AvatarGen where bodies are generated in the canonical pose and then deformed, and the tri-plane representation is used for volume rendering. While explicitly modeling hands and faces is important, it seems like a combination of methods like GNARF which have already been applied to heads, and just putting them in the same model. The additional results with text-driven generation or audio-driven manipulation are very cool, but are not technically novel and are instead a combination of open-source models (and can be applied to any other generative articulable 3D representation).

**Questions:**

1. It is mentioned that the left and right hand are represented with the same set of tri-planes but mirrored. Does this limit the capacity in the generation, i.e. when there are different accessories on either hand?
2. Is truncation used when sampling the generated results from the method? Is it used for sampling for the baseline methods, such as AvatarGen and EVA3D? This should be standard for evaluating qualitative results.
3. I assume this is using the same representation as AvatarGen, but why is it not described in the methodology. Is the representation a NeRF? An SDF? How is volume rendering done?

**Limitations:**

The limitations of the method have not been explicitly addressed in the paper, although they are addressed in the supplementary information. In my opinion, including some of the limitations, especially surrounding the quality of the datasets and SMPL-X estimators (and how this actually affects the quality of the generated results) would be insightful (potentially more-so than the added applications, such as audio-driven acting or text-driven generation).

---

> ### Author Rebuttal · Authors · 2023-08-09
>
> We thank the reviewer for the positive feedback and recognition that 1) our method has SOTA results; 2) our method is ablated well. We respond to each of your comments one by one in what follows.
>
> > **Weakness 1.1**
>
> Thanks for the suggestions, we have computed these metrics for ENARF and EVA3D. The results are shown in Table 1 of our rebuttal PDF file. In general, the results for ENARF and EVA3D are worse than ours because they don't have explicit control over these attributes. However, there is an outlier for ENARF MSE Jaw on UBC dataset. We think the reason is that most of the faces in this dataset have a zero jaw pose. And the images generated by ENARF have a very low resolution and the mouth is always blurry.  In this case, the pose estimator would tend to predict zero jaw poses, which leads to the lowest Jaw MSE.
>
> > **Weakness 1.2**
>
> We visualize more results using different identities and identical mouth open and hand poses. Also, we visualize the random walk results and keep the mouth and hand pose unchanged to verify the consistency of our results. The results can be found on the left of Figure 3 in our rebuttal PDF file.
>
> > **Weakness 2**
>
> We would like to thank the reviewer for the recognition of our results and downstream applications. Although the frameworks mentioned by reviewer could be applied for different 3D generation tasks, these straightforward extensions cannot guarantee high-quality faces or hands simultaneously in one generation model. While we acknowledge the individual components used in our work have been explored in previous studies, combining and adapting them for generative and expressive 3D human modeling is a challenging and non-trivial task. Our sophisticated design not only generates high-quality 3D fully animatable avatars but also achieves clear improvements in face and hands generation/animation. Moreover, our method facilitates various downstream applications, like text-guided synthesis and video/audio-driven animation. Thus, we believe integrating all the important modules coherently and demonstrating their effects through a comprehensive ablation study constitutes a valuable contribution, as also highlighted by Reviewer 3QMX.
>
> > **Question 1**
>
> We conduct an ablation study on the hand Tri-planes, and the results are shown below:
> | Hand Triplanes | FID&#8595; | FID_f&#8595; | FID_h&#8595; | PCK_h&#8593; | Hand&#8595; |
> |-|:-:|:-:|:-:|:-:|:-:|
> | double | 8.32 | 10.12 | 20.53 | 39.64 | 3.27 |
> | single | 5.88 | 10.06 | 19.23 | 38.53 | 3.30 |
> |||||
>
> We can see that using two hand Tri-planes can improve the control ability of hand, but image fidelity decreases. We think the reason is that generator learns to generate an additional hand Tri-plane, which is difficult to optimize. We agree that independent hand Tri-planes can model different accessories, and we observed this during training (Figure 1(c) in our rebuttal PDF file). However, accessories usually have small scales. Compared with the full body image, this point is less important and will not affect the FID score too much. This is confirmed by the FID score for hand.
>
> > **Question 2**
>
> We didn't use any truncation for qualitative results. To make fair comparisons, we disable truncation for all the baseline models. We will point this out in our paper.
>
> > **Question 3**
>
> It is correct, we use the same representation as AvatarGen. We describe this in lines 158-159 of our main text. Our 3D representation is Tri-plane, with SDF as the proxy for geometry. We mention this in line 101 of our main text. Our volume rendering process is identical to NeRF. We will modify the main text to explicitly point these out in the final version.
>
> > **Limitations**
>
> Thanks for the suggestions, we conducted an ablation study to investigate the effects of SMPL-X estimations. We randomly sample a random noise and add it to the clean SMPL-X parameters to make a noisy version of the training dataset. The experiment results are summarized below:
>
> | SMPL-X | FID&#8595; | FID_f&#8595; | FID_h&#8595; | PCK&#8593; | PCK_f&#8593; | PCK_h&#8593; | Exp&#8595; | Shape&#8595; | Jaw&#8595; | Body&#8595; | Hand&#8595; |
> |-----|:------------------:|:-------------:|:----------:|:------------------:|:-------------:|:----------:|:------------------:|:-------------:|:----------:|:-------------:|:----------:|
> | noisy | 8.61 | 10.48 | 19.55 | 62.17 | 90.48 | 31.14 | 6.78 | 3.88 | 7.45 | 1.51 | 3.75 |
> | clean | 5.88 | 10.06   | 19.23 | 65.14 | 91.44 | 38.53 | 5.56 | 3.66 | 6.57 | 1.24 | 3.30 |
> |||||||
>
> As we can see, noisy SMPL-X decreases the model's performance in terms of all the evaluation metrics. It will mainly affect the control ability and full body image fidelity. Therefore, a more precise SMPL-X estimation method may largely improve control ability and full body image quality. We will discuss this in the limitation section and move it from supplementary material to the main text.

---

> > ### Comment · Reviewer_GuK3 · 2023-08-17
> >
> > Thank you for the detailed response to my points. I do not have any additional questions. Overall, I find that the justification for all of the contributions is there, and while none of the components are entirely novel, the system as a whole does enable improved performance, especially for hands and jaw. Thus, I am leaning positively.

---

> > > ### Author Response · Authors · 2023-08-18
> > >
> > > We are glad that all of the reviewer's questions have been answered. Thanks for your valuable time!

---

### Official Review · Reviewer_YASD · 2023-07-02

**Soundness:** 3 good
**Presentation:** 4 excellent
**Contribution:** 3 good
**Rating:** 7
**Confidence:** 5

**Summary:**

The authors proposed novel part-aware sampling and feature parametrization strategies to improve the fidelity of the avatar model, especially for smaller body parts. These techniques s enable the efficient learning of diverse fashion shapes with a focus on the hand and facial details. Through experimental evaluation of several benchmarks, the method surpasses all baselines including the recent work AG3D for a huge margin. The results emphasize the necessity of introducing discriminator techniques. Moreover, the authors demonstrate control with text and audio-generated poses as a possible application of their model.

**Strengths:**

I appreciate the authors' effort in presentation results and getting fair comparisons with baselines.

- Undoubtedly, the proposed technique for discriminative learning is important and interesting - easily transferable to other models, which is clearly well studied
- The authors suggest using separate representations for significant parts - hands, face, and body as a simple technique to drastically improve the capacity of the model
- Comparison presented on three data unrelated to the training corpus.

**Weaknesses:**

- some important architectural choices are not studied. The original idea of sepration face and body doesn’t have cleare explanation or intuition to separate face from the body for features (not in discriminator)
- The ideas lying in the core of the method are quite unpretentious, which is not bad but it decreases the novelty and impactful of the model itself.
- ethical and limitation sections are missing in the main text
- Animations are only shown for synthetic pose sequences. Try to extend it with real sequences (e.g. AMASS), the body movements can improve the credibility of the method.
- No images from the competitor, who utilizes the part-based discriminator as well - AG3D - ask authors to sample more images for you.
- SMPL-based deformation model will affect loose clothing that is not covered in the main text

**Questions:**

- Why do you condition the discriminator for shape and expression (and not pose to handle hands/skeleton deformations or just clear this moment in the main text)? Is it important if so, could you add reference numbers for that?
- Is it possible to compare the methods for human preferences score, since metrics here are not ideal measures? At least a more careful qualitative evaluation can be a good extension it is difficult to see the difference in Figure 4. Is it possible that FID results here do not correlate well with visual ones?
- What if the size of the face and hands tri-plane will be downsampled twice more?
- Does geometry contain any facial attributes related to expression or fingers?
- What is happening with loose clothing since you have quite a simple deformation model?
- How good is this part-based scheme for image inversion - do you have any results of intuition in comparison with EVA3D or AvatarGen.

**Limitations:**

It is difficult to handle wrinkles or accurate textures as well as loose cloth. Moreover, the demonstrated method is less efficient due to the extended representation. Due to the fact that the model is trained without video data, it is not possible to definitively determine the correspondence of pixels can be incorrect. The trained model has a huge bias to fashion poses due to training data that should be taken into account with all conclusions from the model.

---

> ### Author Rebuttal · Authors · 2023-08-09
>
> We thank the reviewer for the positive feedback and constructive suggestions. We respond to each of your comments one by one in what follows.
>
> > **Weakness 1**
>
> Our motivation of separating face and hands from body features is to increase the resolution of Tri-planes and improve the model's capacity for these small-scale regions. This design choice has been confirmed by our ablation studies in Table 3(a) of our main text. When we add two separate Tri-planes for face and hand, we achieve the best results in terms of visual fidelity.
>
> > **Weakness 2**
>
> While we acknowledge the individual components used in our work have been explored in previous studies, combining and adapting them for generative and expressive 3D human modeling is a challenging and non-trivial task. Our carefully designed XAGen not only generates high-quality 3D fully animatable avatars but also achieves clear improvements in face and hands generation/animation. Thus, we believe integrating all the important modules coherently and demonstrating their effects through a comprehensive ablation study constitutes a valuable contribution, as also highlighted by Reviewer 3QMX.
>
> > **Weakness 3**
>
> Thanks for pointing this out, we will move these two sections from supplementary material to the main text.
>
> > **Weakness 4**
>
> We have generated animation results using the sequences sampled from the AMASS dataset. The image results can be found in Figure 1(b) in rebuttal file. We have also sent the link to video results to the AC. In addition, our animation results in initial submission are driven by the sequences from TalkSHOW dataset, which are the tracking results of realistic videos instead of synthetic.
>
> > **Weakness 5**
>
> We have contacted the authors for the AG3D results. The qualitative comparisons can be found in Figure 2(b) in the rebuttal file.
>
> > **Weakness 6**
>
> We agree that SMPL-based deformation will struggle with modeling loose clothing. Loose clothing is a long-standing challenge for 3D human modeling. We believe an advanced human body prior or independent clothing modeling is helpful to alleviate this issue. We will add the discussion of this point into the limitation section in our main text.
>
> > **Question 1**
>
> We choose these designs experimentally. To investigate these choices, we conduct additional ablation studies and the results are shown below:
>
> | Expression | FID_f&#8595; | PCK_f&#8593; | Exp&#8595; |
> |-|:-:|:-:|:-:|
> | w/o | 9.57 | 91.43 | 5.86 |
> | w/ | 10.06 | 91.44 | 5.56  |
> |||||
>
> *Expression:* Although adding expression condition slightly decreases the FID of face, it will improve the control ability in terms of facial expression.
>
> | Body Pose | FID&#8595; | PCK&#8593; | Body&#8595; |
> |-|:-:|:-:|:-:|
> | w/ | 14.23 | 66.69 | 1.00 |
> | w/o | 5.88 | 65.14 | 1.24  |
> |||||
>
> *Body Pose:* It can be observed that conditioning on body pose will significantly affect image quality. Although it can improve the control ability for body pose, the quality decrease is too large.
>
> | Hand Pose | FID_h&#8595; | PCK_h&#8593; | Hand&#8595; |
> |-|:-:|:-:|:-:|
> | w/ | 19.57 | 40.07 | 3.59 |
> | w/o | 19.23 | 38.53 | 3.30  |
> |||||
>
> *Hand Pose:* We can see that conditioning on the hand pose will not affect the visual quality too much. Though the PCK drops, the MSE for hand control is improved. Thus, we skip it to save the computation cost.
>
> > **Question 2**
>
> Yes, we conducted human studies for qualitative comparisons. The results are summarized in Table 2 of our rebuttal PDF file. We can see that our method outperforms baselines in terms of texture and geometry quality on four datasets. The human preference scores are in line with the FID scores reported in Table 1 of our main text.
>
> > **Question 3**
>
> To answer this question, we conducted ablation experiments accordingly. The results are shown in the Table below:
> | Triplanes | FID&#8595; | FID_f&#8595; | FID_h&#8595; |Exp&#8595; | Jaw&#8595; | Hand&#8595; |
> |-|:-:|:-:|:-:|:-:|:-:|:-:|
> | half | 7.05 | 9.59 | 20.33 | 6.17 | 7.03 | 4.11 |
> | full | 5.88 | 10.06 | 19.23 | 5.56 | 6.57 | 3.30 |
> |||||
>
> It can be observed that reducing the size of face and hand Tri-planes twice will cause a performance decrease in terms of full body and hand quality. Also, we can see a performance drop for expression, jaw pose, and hand pose control. These results show that a smaller Tri-plane will decrease the capacity of our generator and decrease the control abilities.
>
> > **Question 4**
>
> We tried our best but still couldn't understand this question. We are glad to further answer this question during discussion period.
>
> > **Question 5**
>
> As shown in the below figure in Figure 6(b) of our main text and the right figure in Figure 1(b) of our rebuttal file, when it comes to loose clothing (e.g., dress), our generator will learn to separate the dress and attach it to the two legs. When the avatar opens its legs, the long dress will also open from the middle. The reason is that our generator adopts the nearest neighbor LBS and each query point will be attached to its nearest SMPL-X vertex in inverse skinning.
>
> > **Question 6**
>
> We report the inversion results in the right above of Figure 3 in rebuttal PDF file. Compared with EVA3D, our result has a better hand and a more detailed geometry. Compared with AvatarGen, we can inverse the T-shirt collar better and we also exhibit a more detailed geometry.
>
> > **Limitations**
>
> Thanks for pointing these limitations out, we will discuss these points in our limitation section and move this section from supplementary material to the main text.

---

> > ### Comment · Reviewer_YASD · 2023-08-21
> >
> > Thank you for addressing all my points with valuable and informative answers. I hope you will add it to the main text. Q4 was about small high-frequency details like wrinkles or fingers. I want to save my original score (accept).

---

> > > ### Author Response · Authors · 2023-08-21
> > >
> > > Thanks for your time and efforts. We are glad that all of the points have been addressed, and we will include the above experiments in the main text of our final version.
> > >
> > > Answer to Question 4: As shown in Figure 2 and Figure 3 (right below) in the rebuttal.pdf, our results show reasonable and detailed geometry for facial expression (clear eyes, nose, and lips) and fingers, while none of the baseline models demonstrate comparable details. We also show dynamic results for opening mouth in the video of our supplementary material. However, wrinkles are more challenging than expressions and fingers. Our method has limitations in modeling wrinkles and we will discuss this in the limitation section. Moving forward, we plan to explore better representations (with stronger capabilities for high-frequency details) and use stronger supervision signals (e.g. normal map) to improve the details.
> > >
> > > We hope this could answer your question.

---

### Official Review · Reviewer_F4xj · 2023-07-05

**Soundness:** 4 excellent
**Presentation:** 2 fair
**Contribution:** 3 good
**Rating:** 6
**Confidence:** 4

**Summary:**

The paper presents a framework for human 3D avatar generation. The whole model is trained on a set of 2D images, thus can support large variations in terms of shape. The framework is based on EG3D with several important modifications:
1) Incorporation of the inverse LBS, that will deform the canonical representation of the object.
2) Modeling hands and face with separate triplane branches.
3) Additional part based discriminators for face and hands.
The paper demonstrates good results, that include modeling of the face and hands motion. Although the motion of these parts are rather weak and contain a lot of artifacts.


POST Rebuttal Summary: The answers in the rebuttal makes sense, in the rebuttal authors provided additional results on sensitivity to SMPL-X errors. To this end I increased the rating by one, and encourage the authors to improve the presentation quality of the method and results.




**Strengths:**

- The paper proposed a meaningful framework that combines several existing ideas into a single unified 3D framework.
- The paper demonstrates good results for the full body modeling.
- The paper contains rigorous evaluation of the different modeling aspects, as well as detailed comparison with sota methods.



**Weaknesses:**

- The quality of presentation is not the best, the method is presented in such a way that it looks like a combination of InsetGan[1] and EG3D. Also visual results presented only on a single dataset, or it is not clear where the results on other datasets.

- Some architectural choices seem questionable. For example why nearest neighbor LBS, was used, instead of the more advanced LBS approximation from HumanNerf[2] or even more advanced SNARF[3]? Also EG3D, because the upsample produces visible inconsistencies in the generated results, why a more advanced training scheme, such as EpiGRAF is not used? The current LBS approximation will struggle with loose clothing that is not well represented with SMPL-X.

- The quality of hands and face is extremely poor. Why it is bad is not analyzed in the paper. Is this a dataset issue? Is it possible to use specific dedicated face/hand datasets in addition to human bodies dataset, and training only face and hand discriminators with them? Or is it an SMPL-X fitting problem? Again is it possible to use more specific hand/face models to improve this aspect?

[1] InsetGAN for Full-Body Image Generation https://arxiv.org/pdf/2203.07293.pdf

[2] HumanNeRF:Free-viewpoint Rendering of Moving People from Monocular Video https://grail.cs.washington.edu/projects/humannerf/

[3] SNARF: Differentiable Forward Skinning for Animating Non-Rigid Neural Implicit Shapes https://arxiv.org/abs/2104.03953

[4] EpiGRAF: Rethinking training of 3D GANs https://arxiv.org/abs/2206.10535




**Questions:**

Questions listed in weaknesses.

**Limitations:**

- Inconsistency in the results.
- Poor quality of the hand and face regions.
- Not very good presentation. Visual results only on a single dataset, no visual comparison with other methods in supplementary videos.

---

> ### Author Rebuttal · Authors · 2023-08-09
>
> We thank the reviewer for the insightful and helpful feedback. We respond to each of your comments one by one in what follows.
>
> > **Weakness 1.1**
>
> First, we agree with the reviewer that our presentation could be further improved. However, there do exist several significant differences between our framework and InsetGan/EG3D:
>
> - InsetGan inserts face/hand images into the full body by optimizing the latent code of each pretrained part GANs to guarantee seamless merging. In contrast, our XAGen does not rely on any optimization or post-processing to merge the images of each part. Instead, our model learns to generate full-body images with plausible faces and hands in one forward pass directly.
>
> - InsetGan is designed for 2D image generation and is not animatable. We believe it is non-trivial to directly apply InsetGan in the animatable 3D avatar generation.
>
> - EG3D is an unconditional 3D-aware GAN model proposed for static face/object modeling, while our model is designed for generating fully-animatable 3D human avatars, which is more challenging due to its high articulation, complex clothes, diverse appearances, and small-scale face and hands. Thus, combining EG3D with InsetGan directly cannot achieve this easily.
>
> - EG3D only synthesizes one single Tri-plane and renders the whole image with one volumetric rendering process. Differently, we propose a multi-part rendering approach together with multi-part discriminators to improve fidelity and controllability for body, face, and hand.
>
> > **Weakness 1.2: visual results**
>
> There may be a misunderstanding because we didn't add explicit captions in Figure 3 of our main text. Indeed, the first row in Figure 3 are generation results on the UBC dataset, and the second row is the results on DeepFashion dataset. We will modify this in the final version of our submission. To evaluate the performance of the other two datasets (SHHQ and MPV), we report more qualitative results in Figure 2 (a) of the rebuttal PDF file.
>
> > **Weakness 2.1: architectural choices**
>
> Thanks for the constructive suggestions. We start from the nearest neighbor LBS to implement our framework, and the experiment results show that this approach is simple yet effective. Since our work is focused on 3D avatar generation instead of a new skinning technique, we didn't further explore advanced techniques such as SNARF or HumanNerf. Moreover, SNARF and HumanNerf were originally designed for single-scene fitting, which may have challenges in terms of generalization ability. Yet, this direction is still worth further exploration, and we will leave this for future work.
>
> > **Weakness 2.2: EpiGRAF**
>
> This question is very insightful. EpiGRAF discards the super-resolution module and proposes a patch-wise training scheme for 3D generation models. Although patch-wise training can reduce the computation cost of each iteration, it also has disadvantages, such as missing global information of the image. This drawback could be problematic for our task because we rely on the body discriminator to guarantee plausibility and increase the consistency between each part (face, body, and hands). Similar to EpiGRAF, EVA3D, one of our baselines, also skips a super-resolution module and synthesizes 512X256 images directly by volume rendering. However, their fidelity and controllability are not comparable with ours. Yet, it is still meaningful to try this advanced training scheme, and we will leave this for future work.
>
> > **Weakness 2.3: loose clothing**
>
> We agree that the LBS approximation will struggle with modeling loose clothing. Loose clothing is a long-standing challenge for 3D human modeling, and it has not been perfectly resolved yet. Even HumanNerf and SNARF with carefully designed skinning techniques and working on single-scene fitting scenario cannot tackle this challenge well. In future works, we may adopt a more advanced human body prior or process clothing independently to address this issue.
>
> > **Weakness 3**
>
> This is a meaningful question. First, we agree that the dataset could be an issue because the quality of the faces and hands cropped from the fashion datasets are less diverse and have a much lower resolution than the commonly used face dataset, such as FFHQ. We believe our method can benefit from a more diverse dataset with higher resolution for face and hands. However, augmenting the training dataset using high-quality face dataset could be challenging because, in this case, there exists a distribution discrepancy between the full body images and face images. Nonetheless, this is an interesting idea and we will explore this direction in future works.
>
> To investigate the SMPL-X fitting problem, we conducted additional ablation studies. We add Gaussian noises into the clean SMPL-X fitting results and train the model again. The results are reported in the Table below:
>
> | SMPL-X | FID&#8595; | FID_f&#8595; | FID_h&#8595; | PCK&#8593; | PCK_f&#8593; | PCK_h&#8593; | Exp&#8595; | Shape&#8595; | Jaw&#8595; | Body&#8595; | Hand&#8595; |
> |-----|:------------------:|:-------------:|:----------:|:------------------:|:-------------:|:----------:|:------------------:|:-------------:|:----------:|:-------------:|:----------:|
> | noisy | 8.61 | 10.48 | 19.55 | 62.17 | 90.48 | 31.14| 6.78 | 3.88 | 7.45 | 1.51 | 3.75 |
> | clean | 5.88 | 10.06   | 19.23 | 65.14 | 91.44 | 38.53| 5.56 | 3.66 | 6.57 | 1.24 | 3.30 |
> |||||||
>
> As we can see, noisy SMPL-X will not affect face and hand image quality too much. It will instead affect control ability and full body quality. Therefore, a more precise or specific SMPL-X/face/hand estimation method may improve control ability more than hands/face quality.
>
> > **Limitations**
>
> We will move the section for limitation discussion from the supplementary material to the main text, and discuss the limitations of our work thoroughly. Please refer to our responses above for the presentation issue and qualitative comparisons on other datasets.

---

### Official Review · Reviewer_3QMX · 2023-07-06

**Soundness:** 4 excellent
**Presentation:** 3 good
**Contribution:** 3 good
**Rating:** 7
**Confidence:** 4

**Summary:**

This work addresses the problem of 3D full body avatar generation, going beyond prior work primarily on more detailed hand and face generation quality and controllability.
The pipeline comprises a 3D-aware GAN where the generator generates tri-plane feature maps from noise vector, followed by a 3D decoder that generates the human geometry represented by SDFs, in a canonical pose. The canonical SDF is then posed (with the help of SMPL-X body model) and rendered to be discriminated about the image discriminators.
The key contributions of the work is a complete pipeline to model the face and hand regions separately, equip them with controllability (e.g. hand pose and facial expression), via the body model, and finally unify them to the full body generation. This effectively addressed the limitation of prior work, namely the compromised hand/face quality due to their relatively small region in most training images. Extensive experiments show the edge of this work over other recent human generators. The authors also demonstrated the usefulness of a full-body avatar generative model in the applications of test/audio-driven avatar synthesis.

**Strengths:**

- This work provides a complete pipeline for controllable 3D avatar generation, where the components in this pipeline are combined in an intuitive and effective way. Speaking of novelty, although the tri-plane-based 3D GANs are common nowadays and the body-model-aided reposing mechanism is also well-known, I can't come up with other work in this line that well fixes the problem of hand/face quality and controllability. Therefore, the full method makes a good technical contribution to the avatar generative modeling.

- Experiment section is strong, and the results are impressive. It incorporates major recent methods on controllable avatar generation and show a clear performance edge in terms of the general coherency of geometry, rendered image fidelity, pose controllability and sharpness on the face/hand region, validating the effectiveness major technical contribution.

- The paper is well-written, being clear and easy to follow. The main paper is mostly self-contained, and the SupMat is also very thorough.

**Weaknesses:**

- The generated geometry has limited resolution, are prone to artifacts, and in most of them, the face/hand region are missing details that are present in the textured renderings. Despite the careful handling of hand/face regions, the geometry of these areas generally show a lower quality than the texture. In other words, although the work targets geometry generation, the model often "covers up" what should be geometric structures with texture.

- Although the generated avatars can indeed be animated (via SMPL-X) as claimed, there seems to be no mechanism that ensures the physical correctness (e.g. semantics of body parts and cloth) or visual plausibility (e.g. pose-aware cloth deformation) of the animated results. For example, when the generated character has self-contact (e.g. Fig 3, lower right "Ours", the subject that touches the face), the contact area are merged as if they are connected (please correct me if wrong); consequently, when animating this subject with novel poses (e.g. arms-open), visual artifacts can appear.

**Questions:**

- The hand, face and body feature maps are generated together in a channel-stack manner. Intuitively, the features of these three different parts are not spatially aligned, and generating them with a same generator might not have other benefits than saving computational cost. Is it investigated whether training separated generators for these 3 parts brings better quality?

- For the face region specifically, AvatarGen also employs a dedicated descriminator, similar to the one used in this work, and yet performs poorly in terms of plausibility of facial detail and expressions. What is the key differentiator in the method that brings such salient improvement?

**Limitations:**

The techinical limitations and potential societal impacts are discussed in detail in the SupMat, I really appreciate that.

---

> ### Author Rebuttal · Authors · 2023-08-09
>
> We thank the reviewer for the positive feedback and recognition that 1) our pipeline is effective, and it makes a good technical contribution to the avatar generative modeling; 2) our results are expressive and show a clear performance edge. We respond to each of your comments one by one in what follows.
>
> > **Weakness 1**
>
> This is a very insightful point. In our pipeline, we apply a super-resolution module to enhance the coarse texture results given by the volumetric rendering, while there is no super-resolution module for geometry. Thus, it is true that the generated geometry has a lower resolution. Based on our current pipeline, there may exist two possible solutions. First, we increase the volume rendering resolution for geometry and show the comparison results in the right below figure in Figure 3 of the rebuttal PDF file. It can be observed that, with an increasing rendering resolution, we can see more geometric details on the face and dress. It demonstrates that our model can improve geometry by using a larger rendering resolution. Yet, this technique can not fully address this issue, and the second solution is to apply a real upsampling on the 3D representation to enhance the details of geometry. We would leave this for future work.
>
> > **Weakness 2**
>
> - To study the issue of contact area, we rotate the samples shown in Fig 3 in our main paper and render more results from multiple directions. As shown in Figure 1 (a) of our rebuttal file, we rotate the avatar by 50, 110, 220, and 330 degrees respectively. The results demonstrate that the hand is not connected or merged with the face.
>
> - It is true that when animating the subject with novel poses like arms-open, we could observe visual artifacts. The main reason for this phenomenon is that our training dataset contains only fashion data, whereas arms-open poses are less frequent in the dataset. We believe a more diverse dataset with in-the-wild poses would help alleviate this issue.
>
> - It is our limitation that our model does not incorporate physical correctness to improve visual plausibility. We agree that these constraints are helpful, and we will incorporate physical/penetration constraints to further improve the animation results in future works.
>
> > **Question 1**
>
> This is a good question. First, we have some design choices to help align the features of different parts: (1) As introduced in our main paper (lines 163-167) and Appendix (Section 1.1), we compute the feature for the query points which are located in the overlap regions from two related Tri-planes, and then composite them to increase the transition smoothness and plausibility of the full body image. (2) Our body discriminator is trained on full body images to critique the generation result. It supervises the learning of full body images and the gradients can be backpropagated to not only the body but also face and hand Tri-planes to increase generation quality and help align the features spatially.
>
> To study the advantages of using a shared generator branch to generate multi-part features, we conduct additional ablation study experiments. We use separated generator branches for hand, face, and body respectively. The results are summarized in the Table below:
>
> | Generator | FID&#8595; | FID_f&#8595; | FID_h&#8595; |
> |-----|:------------------:|:-------------:|:----------:|
> | separated | 7.65 | 12.20 | 21.03 |
> | shared | 5.88 | 10.06   | 19.23  |
> |||||||
>
> The results show that using separated generators cannot improve generation quality. We think the reason would be the redundancy in separated generators. The redundancy increases the computation cost and hinders the optimization of the generator.
>
> > **Question 2**
>
> The differences between ours and AvatarGen are:
>
> - The rendering process of face is different. AvatarGen crops face images from synthesized full body images whereas ours use face camera poses to render the face images directly from face Tri-planes. Compared with AvatarGen, our independent rendering process can disentangle the learning of face and body, which reduces the training difficulty on 2D image datasets.
> - Our multi-part discriminators have stronger supervision on face than AvatarGen. In our pipeline, we have face discriminator and full body discriminator. We, therefore, compute adversarial loss terms for both full body and face. And both full body discriminator and face discriminator critique the face regions and supervise the synthesis of faces, which further enhances face quality.
> - Our face discriminator is conditioned on facial expressions while AvatarGen does not use such conditions. Thus, our discriminator leverages the prior knowledge provided by SMPL-X to supervise the training of generator. It can help enhance facial details (i.e., expressions) of the generated faces.
>
> The above three reasons work together to bring a large improvement in the face synthesis of our pipeline.

---

> > ### Comment · Reviewer_3QMX · 2023-08-18
> >
> > I thank the authors for the detailed response. The rebuttal has addressed my concerns. I also share a similar opinion with other reviewers in several aspects.  Indeed the technical contribution can roughly be seen as a insightful combination of existing techniques, but the results are nice, surpass the SOTAs, and the study is comprehensive. Therefore I'd keep my original recommendation for acceptance. As mentioned by other reviewers, it would be helpful to include at least some of the limitations in the main text. Please include the extended experiments and illustrations from the rebuttal in the final version. Great job!

---

> > > ### Author Response · Authors · 2023-08-18
> > >
> > > Thanks for the reviewer's valuable comment. We are glad that we have addressed the concerns. We will add the limitation section, extended experiments, and illustrations in the main text of our final version.

---

### Author Rebuttal · Authors · 2023-08-09

We want to thank all the reviewers for their constructive and insightful feedback. We appreciate the reviewers' time and efforts spent on our submission.

Please check our rebuttal PDF files uploaded here for the additional figures and tables. We have sent the video results on AMASS data to AC following the official instructions.

Please check the comments for each review, and feel free to ask if you have any questions about the rebuttal. We are glad to have further discussions with all the reviewers.

---

### Decision · Program_Chairs · 2023-09-21

**Decision:**

Accept (poster)

**Comment:**

The paper proposes a 3D-aware GAN generative model for controllable avatar creation purely from 2D images in the wild. The main contributions of the work are in proposing a unified framework for animatable avatar creation with significantly improved quality and controllability for the face and hand regions versus the state of the art. The proposed method, although comprises of a combination of known components, shows strong improvements in quality over the existing approaches. All novel aspects of the work are thoroughly ablated and the presentation quality of the paper is good.

All reviewers' concerns were sufficiently addressed during the rebuttal phase. There is consensus among the reviewers to accept the paper (3A, WA, BA).

The AC recommends acceptance. Congratulations!

The authors are strongly encouraged to incorporate the changes that they've promised in the rebuttal into the final manuscript.